



**High atmospheric oxidation capacity drives wintertime nitrate pollution in the eastern**
**Yangtze River Delta of China**
Han Zang[1], Yue Zhao[1,*], Juntao Huo[2], Qianbiao Zhao[2], Qingyan Fu[2], Yusen Duan[2,*], Jingyuan Shao[3],
Cheng Huang[4], Jingyu An[4], Likun Xue[5], Ziyue Li[1], Chenxi Li[1], Huayun Xiao[1]
[1]School of Environmental Science and Engineering, Shanghai Jiao Tong University, Shanghai, 200240,
China
[2]Shanghai Environmental Monitoring Center, Shanghai 200235, China
[3]College of Flight Technology, Civil Aviation University of China, Tianjin 300300, China
[4]Shanghai Academy of Environmental Sciences, Shanghai 200233, China
[5]Environment Research Institute, Shandong University, Qingdao, Shandong, 266237, China
*Correspondence: Yue Zhao (yuezhao20@sjtu.edu.cn); Yusen Duan (duanys@yeah.net)



**Abstract**
Nitrate aerosol plays an increasingly important role in wintertime haze pollution in China. Despite
intensive research on the wintertime nitrate chemistry in recent years, quantitative constraints on
the formation mechanisms of nitrate aerosol in the Yangtze River Delta (YRD), one of the most
developed and densely populated regions in eastern China, remain inadequate. In this study, we
identify the major nitrate formation pathways and their key controlling factors during the winter
haze pollution period in the eastern YRD using two-year (2018-2019) field observations and
detailed observation-constrained model simulations. We find that the high atmospheric oxidation
capacity, coupled with high aerosol liquid water content (ALWC), made both the heterogeneous
hydrolysis of dinitrogen pentoxide ($N_2O_5$) and the gas-phase OH oxidation of nitrogen dioxide ($NO_2$)
important pathways for wintertime nitrate formation in this region, with contribution percentages of
69% and 29% in urban areas and 63% and 35% in suburban areas, respectively. We further find that
the gas-to-particle partitioning of nitric acid ($HNO_3$) was very efficient so that the rate-determining
step in the overall formation process of nitrate aerosol was the oxidation of $NO_x$ to $HNO_3$ through
both heterogeneous and gas-phase processes. The atmospheric oxidation capacity (i.e., the
availability of $O_3$ and OH radicals) was the key factor controlling the production rate of $HNO_3$ from
both processes. During the COVID-19 lockdown (January-February 2020), the enhanced
atmospheric oxidation capacity greatly promoted the oxidation of $NO_x$ to nitrate and hence
weakened the response of nitrate aerosol to the emission reductions in urban areas. Our study sheds
light on the detailed formation mechanisms of wintertime nitrate aerosol in the eastern YRD and
highlights the demand for the synergetic regulation of atmospheric oxidation capacity and $NO_x$
emissions to mitigate wintertime nitrate and haze pollution in eastern China.



## 1. Introduction

Atmospheric fine particulate matter (PM$_{2.5}$) has profound impacts on air quality, climate, and public
health (Huang et al., 2014; Wang et al., 2014; Lelieveld et al., 2015; von Schneidemesser et al.,
2015). Over the past decades, China has encountered severe PM$_{2.5}$ pollution due to the rapid
urbanization and industrialization (Huang et al., 2014; Zhang and Cao, 2015; Tao et al., 2017; Peng
et al., 2021). To tackle severe air pollution, Chinese government has implemented active clean air
policies such as the "Action Plan for Air Pollution Prevention and Control" in recent years. As a
result, anthropogenic emissions of major air pollutants such as sulfur dioxide (SO$_2$), nitrogen oxides
(NO$_x$), and primary PM have declined dramatically and the nationwide PM$_{2.5}$ air quality have
improved significantly (Shao et al., 2018; Zheng et al., 2018; Ding et al., 2019; Zhang et al., 2019).
In addition, with the emission reduction of primary PM, secondary aerosol has become the most
important component of PM$_{2.5}$ (Shao et al., 2018; Ding et al., 2019; Peng et al., 2021).
Secondary inorganic aerosol consisting mainly of nitrate, sulfate, and ammonium (SNA),
contributed to 30-60% of the PM$_{2.5}$ mass in China (Hua et al., 2015; Tao et al., 2017; Ye et al., 2017;
Wang et al., 2018; Fu et al., 2020; Lin et al., 2020). During the pollution episodes, the proportion of
SNA to PM$_{2.5}$ could exceed 50% (Tao et al., 2017; Liu et al., 2020; Peng et al., 2021). Before 2013,
sulfate was often found to be the most abundant component of PM$_{2.5}$ in Chinese cities (Zhao et al.,
2013; Huang et al., 2014; Kong et al., 2014; Xie et al., 2015; Tao et al., 2017). However, with the
implementation of stringent clean air policies, anthropogenic emissions of SO$_2$ in China had
dropped by 59% from 2013 to 2017, while NO$_x$ emissions decreased only by 21% during the same
period (Zheng et al., 2018). Consequently, sulfate aerosol concentration has decreased dramatically
nationwide since 2013, but wintertime nitrate concentration has not decreased much (Ding et al.,
2019; Li et al., 2019a; Xu et al., 2019; Fu et al., 2020; Wang et al., 2020b); nitrate has become an
increasingly important component of PM$_{2.5}$ in most regions of China during winter (Ye et al., 2017;
Yun et al., 2018; Li et al., 2019a; Xu et al., 2019; Chen et al., 2020; Fu et al., 2020; Kong et al.,
2020; Lin et al., 2020; Xie et al., 2020; Zhai et al., 2021; Zhang et al., 2021). The high loading of
nitrate has been considered playing an important role in winter haze pollution (Wen et al., 2015;
Sun et al., 2018). Therefore, identifying the major nitrate formation pathways and their controlling
factors during haze events is of great importance for developing effective particulate pollution
mitigation policies in China.
In polluted regions, the nitrate aerosol arises mainly from two pathways: (1) the gas-phase oxidation
of nitrogen dioxide (NO$_2$) by OH radicals producing nitric acid (HNO$_3$) (Calvert and Stockwell,
1983) and (2) the heterogeneous hydrolysis of dinitrogen pentoxide (N$_2$O$_5$) that was produced from
the reaction of NO$_2$ with nitrate (NO$_3$) radicals, on aqueous aerosols (Bertram and Thornton, 2009;
Bertram et al., 2009; Wagner et al., 2013; McDuffie et al., 2019). The gas-phase OH + NO$_2$ pathway
primarily occurs during the daytime and is mainly influenced by the atmospheric oxidation capacity
despite the NO$_2$ concentration (Chen et al., 2020; Fu et al., 2020). The heterogeneous formation of
nitrate via N$_2$O$_5$ hydrolysis is greatly affected by aerosol liquid water content (ALWC) and the





production of $N_2O_5$ (Alexander et al., 2020; Lin et al., 2020; Wang et al., 2020b). As a result, this
heterogeneous pathway is generally weak during the daytime because of the fast photolysis of $NO_3$
radicals or titration by NO (Wayne et al., 1991; Brown and Stutz, 2012), which inhibit $N_2O_5$
production. However, it could be the dominant pathway for nitrate formation during the nighttime
(Wang et al., 2017; McDuffie et al., 2019), where $N_2O_5$ can be produced more efficiently and its
hydrolysis is favored by the high relative humidity (or ALWC).
There have been a number of field studies on the pollution characteristics and formation
mechanisms of nitrate aerosol during haze events in China over the past decades (Tao et al., 2016;
Li et al., 2018; Sun et al., 2018; Wen et al., 2018; Ding et al., 2019; Ye et al., 2019; Chen et al., 2020;
Fu et al., 2020; Lin et al., 2020; Wang et al., 2020b; Zhao et al., 2020a; Chan et al., 2021). However,
most of these studies were carried out in the North China Plain (NCP) (Li et al., 2018; Wen et al.,
2018; Chen et al., 2020; Fu et al., 2020; Wang et al., 2020b; Chan et al., 2021). Earlier studies
suggested that the nitrate formation during the pollution episodes in this region was mainly
attributed to the heterogeneous hydrolysis of $N_2O_5$ (Su et al., 2017; Wang et al., 2017; He et al.,
2018; Li et al., 2018). However, recent studies showed that the gas-phase OH + $NO_2$ process has
become more important, and sometimes this process was even the dominant pathway for nitrate
formation (Chen et al., 2020; Fu et al., 2020). The Yangtze River Delta (YRD) in eastern China is
one of the most developed regions in China (Ding et al., 2013). The wintertime $O_3$ concentration is
relatively high in this region, with an average of ~20 ppb, and sometimes could even reach 75 ppb
(Li et al., 2019c; Ye et al., 2019; Zhao et al., 2020a), which is significantly higher than that (average:
6-16 ppb) in the NCP region (Li et al., 2019a; Duan et al., 2020; Liu et al., 2020). Furthermore, the
relative humidity (RH) in this region is also high, with the average winter RH ranging from 63% to
71% (Tao et al., 2016; Shen et al., 2020; Yu et al., 2020b), which was also significantly higher than
the average RH (20-40%) in the NCP region (Fang et al., 2019; Li et al., 2019a; Huang et al., 2020;
Xie et al., 2020). The high atmospheric oxidation capacity, coupled with the high RH that led to
high ALWC, would favor the production of secondary aerosol (Peng et al., 2021).
Haze pollution events frequently occurred in the YRD during winter (Hua et al., 2015; Sun et al.,
2018; Ding et al., 2019). Although there have been many studies on the pollution characteristics of
nitrate and $PM_{2.5}$ in this region (Tao et al., 2016; Sun et al., 2018; Chen et al., 2019; Ding et al.,
2019; Ye et al., 2019; Lin et al., 2020; Shen et al., 2020), only a few studies have focused on the
nitrate formation mechanisms. It has been reported that the heterogeneous hydrolysis of $N_2O_5$
contributed dominantly to nitrate formation in the western YRD (Sun et al., 2018), and its
production rate could be 5 times higher than that of the gas-phase OH + $NO_2$ process during severe
haze pollution events (Lin et al., 2020). In contrast, some other studies have qualitatively pointed
out that the gas-phase OH + $NO_2$ reaction was an important formation pathway of nitrate in the
eastern YRD, though the heterogeneous hydrolysis of $N_2O_5$ during the nighttime also contributed
(Ye et al., 2019; Zhao et al., 2020a). Overall, quantitative constraints on the detailed formation
mechanisms of wintertime nitrate aerosol in the YRD region remain limited. The relative



contribution of different nitrate formation pathways and their controlling factors are still unclear.

In this study, we conducted hourly measurements of nitrate and associated particulate and gaseous
air pollutants at an urban site and a regional site in the eastern YRD during winter in 2018 and 2019,
aiming to clarify the nitrate formation mechanisms during winter. An observation-constrained box
model using the detailed Master Chemical Mechanism (MCM v3.3.1) updated with the state-of-the-
art heterogeneous chemistry of $N_2O_5$, $NO_2$, and particulate nitrate was employed to quantitatively
identify the major reaction pathways and key controlling factors for wintertime nitrate aerosol
formation in this region. This study will help to understand the nitrate aerosol chemistry in the
eastern YRD and develop effective strategies to mitigate secondary aerosol pollution in this densely
populated region.

**2. Materials and methods**
**2.1 Observation sites and instrumentation**
$PM_{2.5}$ and its chemical composition, inorganic gases, volatile organic compounds (VOCs), and
meteorological parameters were continuously measured at a regional site (Qingpu) and an urban site
(Pudong) in Shanghai from December 1 to February 12 in both 2018 and 2019. The Qingpu site
(120.989 °E, 31.097 °W) is a suburban site (see Fig. 1), located near the Dianshan Lake and
surrounded by the residential areas and vegetation, and about 46 km away from the urban Shanghai.
Besides, the Qingpu site is located at the junction of Shanghai, Jiangsu, and Zhejiang province and
is a typical regional site in the eastern YRD. The instruments at this site were on the rooftop of a 10
m tall building. The Pudong site (121.533 °E, 31.228 °W) is an urban site located near the Century
Avenue with heavy traffic, and it is only ~3 km from the business center Lujiazui. The instruments
at this site were located on the roof of a 20 m tall building. The eastern YRD region is affected by
the subtropical monsoon climate, dominated by the northwest and northeast winds in winter.

The measurements at the two sites were conducted hourly. The $PM_{2.5}$ mass concentration was
measured by a Tapered Element Oscillating Microbalance combined with Filter Dynamic
Measurement System (TEOM-FDMS, TEOM 1405-F, Thermo Fisher Scientific, USA.). Water-
soluble ions including $NO_3^-$, $SO_4^{2-}$, $NH_4^+$, $Cl^-$, $Na^+$, $Ca^{2+}$, and $Mg^{2+}$ were measured using an online
Monitor for Aerosol and Gases (MARGA, ADI 2080, Applikon Analytical B.B.Corp., Netherlands).
Organic carbon (OC) and elemental carbon (EC) were measured by a semi-continuous OC/EC
analyzer (Model 4, Sunset Laboratory Inc., USA), and a denuder was installed before analyzer to
avoid the disturbance of organic vapors. The surface area and volume concentrations of aerosol
particles were measured using a scanning mobility particle sizer (SMPS, TSI, USA, which consists
of a 3080 electrostatic classifier, a 3081A different mobility analyzer, and a 3787 condensation
particle counter) and an aerodynamic particle sizer (APS 3321, TSI, USA). The combination of
SMPS and APS was able to cover the particle size range from 13.6 nm to 10 μm. Considering that
the Pudong sampling site lacks the data of aerosol volume and surface area concentrations, we
performed a linear fit between the aerosol surface/volume and $PM_{2.5}$ mass concentration at the





Qingpu site (see Figure S1 in the supplement), and predicted the values for the Pudong site based
on such a linear fit and the measured $PM_{2.5}$ mass concentration. The surface/volume concentrations
of dry aerosol particles measured by SMPS and APS were corrected to the ambient RH based on an
empirical composition-kappa function and the kappa–Köhler function (see details in Section S1 of
the Supplement). The $O_3$, $NO_x$, and $SO_2$ were measured by an Ozone, $NO_x$, and $SO_2$ analyzer (Model
49i, 42i, and 43i, Thermo Fisher Scientific, USA), respectively. A total of 56 VOCs were measured
using gas chromatography equipped with a flame ionization detector (GC-FID, Chromatotec
A11000/A21022 at the Qingpu site and PerkinElmer Clarus 580 at the Pudong site). Meteorological
parameters including temperature, RH, pressure, wind speed and direction were measured by a
meteorological transducer (WXT520, Vaisala Ltd., Finland).

**2.2 Estimation of aerosol liquid water content and pH**
The ISORROPIA-II thermodynamic model was used to calculate aerosol pH and ALWC
(Fountoukis and Nenes, 2007). The water-soluble inorganic ion concentrations, along with RH and
temperature, were used as the model input. The model was run in the forward mode, which would
give a more accurate estimation of aerosol pH than using the reverse mode with only particulate
inorganic ions as the model input (Guo et al., 2015; Hennigan et al., 2015). Besides, considering the
relatively high RH in eastern YRD, we selected the metastable state for aerosol in this study.
ISORROPIA-II calculated the equilibrium concentrations of particle hydronium ions ($H^+_{air}$, μg m$^{-3}$)
and ALWC (μg m$^{-3}$) in per air volume. Then the aerosol pH can be derived by the following equation:
$$pH = -\log_{10}(H^+_{aq}) = -\log_{10}\frac{1000H^+_{air}}{ALWC} \qquad (1)$$
Where $H^+_{aq}$ is the concentration of hydronium ions in aqueous aerosol (mol L$^{-1}$). It should be
mentioned that when the RH was extremely high (> 95%), a slight deviation in measured RH would
cause significant uncertainty in the estimation of ALWC. Therefore, we only considered the data
with the RH below 95% in the further analysis.

**2.3 Observation-constrained model simulation**
The Framework for 0-D Atmospheric Modeling (F0AM v3.1) (Wolfe et al., 2016) employing the
MCM v3.3.1 (Jenkin et al., 2015) was used to simulate the formation of nitrate in the pollution
events during the whole observation period. Figure 2 summarizes the formation pathways of $HNO_3$
in the atmosphere (Alexander et al., 2020; Chan et al., 2021). In the model, we considered the
reaction pathways including heterogeneous hydrolysis of $N_2O_5$ (R3) and $NO_2$ (R8), gas-phase OH
+ $NO_2$ (R7), $NO_3$ radical oxidation of VOCs (R5), and reaction of NO with hydroperoxy ($HO_2$)
radicals (R2), which together contributed to 88% of $HNO_3$ formation in the global troposphere
(Alexander et al., 2020). The model did not include the hydrolysis of $NO_3$ radicals and organic
nitrate (R1, R4, and R6), as well as the reaction of $NO_2$ with halogen oxide species (R9). However,
these pathways only had a small contribution to the production of $HNO_3$ (Alexander et al., 2020).
Therefore, they would not significantly affect the model results in this study.





The default MCMv3.3.1 does not consider the heterogeneous hydrolysis of $N_2O_5$ in detail and the
heterogeneous production of nitrous acid (HONO), an important precursor of OH radicals in the
polluted atmosphere. Therefore, we parameterized these processes in the model based on recent
advances in these processes. The rate of the heterogeneous hydrolysis of $N_2O_5$ on aqueous aerosols
($k_3$) could be calculated by eq. 2 when ignoring the gas-phase diffusion limitation:
$$k_3 = \frac{\gamma\,N_2O_5 \cdot c \cdot S_a}{4} \tag{2}$$
where $\gamma N_2O_5$ is the uptake coefficient of $N_2O_5$, defined as the probability of removal of $N_2O_5$ per
collision with the wet aerosol surface; $c$ is the mean molecular speed of $N_2O_5$; $S_a$ is the measured
aerosol surface area concentration. In this study, we employed an observation-based empirical
parameterization of $\gamma N_2O_5$, which provided a reasonable representation of the $PM_{2.5}$ reactivity
toward $N_2O_5$ at different Chinese sites, according to a recent study (Yu et al., 2020a):
$$\gamma N_2O_5 = \frac{4}{c}\frac{V_a}{S_a}\,K_H \times 3.0 \times 10^4 \times [H_2O] \times (1 - \frac{1}{\left(0.033 \times \frac{[H_2O]}{[NO_3^-]}\right) + 1 + \left(3.4 \times \frac{[Cl^-]}{[NO_3^-]}\right)}) \tag{3}$$
where $V_a$ is the measured aerosol volume concentration; $K_H$ is the Henry's law coefficient of $N_2O_5$,
with a value of 51 M atm$^{-1}$ (Bertram and Thornton, 2009); and [$H_2O$], [$NO_3^-$], and [$Cl^-$] are the
molarity of water, nitrate, and chloride in aerosol, respectively.

The heterogeneous hydrolysis of $N_2O_5$ on aqueous aerosols could form $HNO_3$ and/or nitryl chloride
($ClNO_2$) (see R3), with their yields (i.e., the value of $\Phi$, ranging between 0 and1) depending on the
$H_2O$ and $Cl^-$ content in the aerosol (Bertram and Thornton, 2009; Yu et al., 2020a). In this study, the
yield of $HNO_3$ ($\Phi_{HNO3}$) was estimated from eq. 4 (Bertram and Thornton, 2009; Yu et al., 2020a):
$$N_2O_5 + Aerosol \rightarrow 2\Phi\,HNO_3 + 2(1-\Phi)\,ClNO_2 \tag{R3}$$
$$\Phi_{HNO3} = 1 - 1/(1 + \frac{[H_2O]}{105 \times [Cl^-]}) \tag{4}$$

Photolysis of HONO was shown to contribute 20-92% of the production of OH radicals during
winter haze pollution events in China (Tan et al., 2017; Slater et al., 2020; Xue et al., 2020). Here,
on the basis of previous studies (Lee and Schwartz, 1983; Kleffmann et al., 1998; Kurtenbach et al.,
2001; Wong et al., 2011; Wong et al., 2013; Han et al., 2016; Ye et al., 2016; Liu et al., 2017; Trinh
et al., 2017; Romer et al., 2018; Zare et al., 2018; Liu et al., 2019; Wang et al., 2020a; Xue et al.,
2020), we parameterized the major heterogeneous production pathways of HONO and its dry
deposition to estimate the HONO budget during the pollution episodes. The added mechanisms are
summarized in Table 1. A detailed description of the parameterization is provided in the Supplement
(Section S2). Considering that there remain significant uncertainties in the key parameters (i.e., the
uptake coefficient of $NO_2$ on aerosol or ground surfaces, EF, and HONO emission ratios) of the
heterogeneous HONO formation pathways and its direct emissions as listed in Table 1, we
performed the sensitivity analyses for these parameters to evaluate their influences on the model





results.

In addition, we included the dry deposition of $HNO_3$ with a velocity ($v HNO_3$) of 0.0175 m s$^{-1}$ (Liu
et al., 2019). The rate constant of the deposition was then calculated using eq. 5:
$$k_{dep} = \frac{v HNO_3}{PBL} \tag{5}$$
where PBL is the planetary boundary layer height. We also considered the dilution of all other
species via deposition, entrainment, etc. using a highly simplified parameterization:
$$\frac{d[X]}{dt} = - k_{dil} ([X]-[X]_{bkg}) \tag{6}$$
where $k_{dil}$ is the first-order dilution rate constant; $[X]_{bkg}$ is a fixed background concentration of
pollutants. Here, a typical dilution lifetime of one day was assumed, i.e., $k_{dil}$ = 1/86400 s$^{-1}$. As the
species background concentration was unknown, $[X]_{bkg}$ was set to 0 for simplicity.

In the model, the $j$ values of various gaseous species were calculated using the default MCMv3.3.1
parameterization with input of the solar zenith angle at the observations sites and scaled by the ratio
of measured to calculated $jNO_2$ values. The observed pollutant concentrations and meteorological
parameters were used as the model input, which were updated hourly (one model step) using the
observation data and held constant during each model step, except for the observed concentrations
of NO and $NO_2$ (the sum of NO and $NO_2$ concentrations was also constrained by the observation).

**3. Results and Discussion**
**3.1 Overview of pollution characteristics during winter**
Table 2 shows the overall pollution conditions of the two observation sites in winter 2018 and 2019.
The average $PM_{2.5}$ concentration increased by 17-21% in 2019 compared to that in 2018.
Accordingly, nitrate concentration also increased by 11-14% in 2019. The $O_3$ concentration was
slightly higher in 2019 than in 2018, consistent with increased atmospheric oxidation capacity in
recent years (Lu et al., 2018; Li et al., 2019b; Liu and Wang, 2020; Yang et al., 2020). In the two
years, both of the $PM_{2.5}$ and nitrate concentrations at the Qingpu site were higher than those at the
Pudong site. As mentioned above, the Qingpu site is at the junction of Shanghai, Jiangsu, and
Zhejiang, so it is more easily influenced by the transport of air pollutants from Jiangsu, which is
usually more polluted than Shanghai. Besides, the average temperature at the Qingpu site was also
slightly lower than that at the Pudong site, which might to some extent favor the gas-to-particle
partitioning of $HNO_3$. Notably, the average RH was as high as 80% during the observation period,
which was significantly higher than that (63%) recorded in 2016 (Tao et al., 2016). In particular, the
RH exceeded 90% for more than one third of the days during the observation period.

Taking the Pudong site in 2019 as an example, we analyzed the time series of $PM_{2.5}$, nitrate, and
other related parameters and presented the results in Figure 3 (Time series of the pollutants at the
Qingpu site can be seen in Section S3 and Figure S2). $PM_{2.5}$ pollution events occurred frequently in



the eastern YRD during winter. During the observation period, the PM$_{2.5}$ concentration exceeded 75
µg m$^{-3}$ for 34 days and 150 µg m$^{-3}$ for 6 days. During the pollution episodes (PM$_{2.5}$ > 75 µg m$^{-3}$),
nitrate had become the most important component of PM$_{2.5}$, and its concentration was a factor of
2.2 higher than that of sulfate. In winter, the emission of NO$_x$ was obviously high. During the periods
with high nitrate concentration, the NO$_x$ concentration always exceeded 100 ppb. The O$_3$
concentration was also at a relatively high level, with a maximum value of 60 ppb and an average
of 22 ppb, which was much higher than the wintertime average O$_3$ concentration (6-16 ppb) in the
NCP (Li et al., 2019a; Duan et al., 2020; Liu et al., 2020). The concentration of odd oxygen
(O$_x$=O$_3$+NO$_2$) ranged between 20-83 ppb with an average of 44 ppb, indicating a relatively high
atmospheric oxidation capacity in the eastern YRD during winter. Consistently, the nitrogen
oxidation ratio (NOR, NOR = NO$_3^-$/(NO$_3^-$ + NO$_2$)) was up to 0.51, suggesting a high degree of
atmospheric oxidation. Meanwhile, the high atmospheric RH in the eastern YRD led to a high
ALWC. During the high nitrate periods, the ALWC was often at its peak and could exceed 200 µg
m$^{-3}$ on rainy or haze-foggy days. Such a high ALWC level would have an important impact on the
nitrate formation. Notably, the NO$_x$ concentration dropped sharply on 23 January and kept at a low
level until the end of the observation (12 February, 2020). This is mainly a result of marked emission
reductions during the COVID-19 lockdown. Such an emission reduction had a complicated
influence on the nitrate formation chemistry, which will be discussed in detail in Section 3.5.

Figure 4 shows the mass ratio of nitrate to PM$_{2.5}$ as a function of the PM$_{2.5}$ concentration and ALWC
at Qingpu and Pudong sites in 2018 and 2019. The ratio of nitrate to PM$_{2.5}$ increased with increasing
PM$_{2.5}$ concentration. When the PM$_{2.5}$ concentration was above 75 µg m$^{-3}$, the average mass fraction
of nitrate was more than 30%. In addition, the nitrate formation rate was much higher than that of
sulfate and ammonium during PM$_{2.5}$ pollution episodes, as indicated by the slope of nitrate vs. PM$_{2.5}$
that was twice that of the other two ions (see Figure S3). These results indicate that the formation
of nitrate played a driving role in the formation of PM$_{2.5}$ pollution. In general, when the ALWC was
high, the nitrate concentration was also at a high level. On one hand, ALWC could promote the
nitrate formation by favoring the heterogeneous hydrolysis of N$_2$O$_5$ and the gas-to-particle
partitioning of HNO$_3$. On the other hand, the increase in nitrate concentration could enhance the
hygroscopicity of PM$_{2.5}$, leading to an increase in ALWC, which would further promote the nitrate
formation (Wang et al., 2020b). It is worth noting that, when PM$_{2.5}$ < 100 µg m$^{-3}$, the mass ratio of
NO$_3^-$ to PM$_{2.5}$ increased rapidly with rising PM$_{2.5}$ concentration, but when the PM$_{2.5}$ concentration
exceeded 100 µg m$^{-3}$, the ratio reached a plateau. This might be due to the fact that when the PM$_{2.5}$
concentration increased to a certain level, the formation process of other components may also speed
up, causing the nitrate proportion to stay basically constant.

**3.2 Gas-to-particle partitioning of nitrate**
The gas-to-particle partitioning of nitrate determines the sensitivity of particulate nitrate formation
to the production of HNO$_3$. Figure 5 shows the particulate nitrate concentration (measured) and its
fraction to total nitrate ($\varepsilon$HNO$_3$, $\varepsilon$HNO$_3$ = NO$_3^-$/(NO$_3^-$ + HNO$_3$), predicted by ISORROPIA-II) as a



function of ALWC and aerosol pH. In order to avoid the influence of rainy and foggy days during
the observation period which could lead to the abnormal high ALWC, we only used the data with
RH below 95% for analysis. Obviously, ALWC promoted the formation of particulate nitrate, but
such a promoting effect varied greatly under different aerosol pH (top panel in Figures 5a-d). As the
pH increased, the slope of nitrate vs. ALWC also increased significantly, indicating a stronger
promoting effect. ALWC plays a dual role in the formation of nitrate aerosol: it can promote the
heterogeneous formation of nitrate, e.g., via $N_2O_5$ hydrolysis, by providing more reaction medium
and decreasing the kinetic limitation (Mozurkewich and Calvert, 1988; Bertram and Thornton, 2009;
Wang et al., 2020b); the ALWC can also promote the gas-to-particle partitioning of $HNO_3$. The
different promoting effect of ALWC under different aerosol pH is mainly due to the fact that pH can
significantly influence the gas-to-particle partitioning of $HNO_3$. As shown in Figures 5a-d (bottom
panel), when the aerosol pH was low, the gas-to-particle partitioning of $HNO_3$ was inhibited, with
the value of $\varepsilon HNO_3$ basically below 0.6 at pH < 2. Under these conditions, the increase of particulate
nitrate concentration would require more ALWC. When the pH increased, the inhibition effect of
pH on the gas-to-particle partitioning of $HNO_3$ was weakened. When the pH was higher than 2.5,
the nitrate was almost in the particle phase ($\varepsilon HNO_3=1$). As a result, the increase of ALWC would
rapidly promote the nitrate formation, particularly when ALWC was at a low level. It is important
to point out that during the whole observation period, the values of $\varepsilon HNO_3$ were larger than 0.9 for
90% of time when the $PM_{2.5}$ concentration was higher than 75 μg m$^{-3}$ (see Figure S4). This indicates
that the gas-to-particle partitioning of $HNO_3$ was very efficient and not a limiting factor for
particulate nitrate formation during the pollution episodes. The gas-to-particle partitioning of $HNO_3$
was also efficient in the NCP region, and its average $\varepsilon HNO_3$ could reach 100% during the haze
pollution period (Guo et al., 2018; Li et al., 2019a). However, the average $\varepsilon HNO_3$ in the northeastern
United States during winter was only 39% (Guo et al., 2018), this might be due to the relatively
lower pH in this region ($0.8 \pm 1.0$) (Guo et al., 2016), which inhibited the gas-to-particle partitioning.

**3.3 Observational constraints on the nitrate formation mechanism**
The nitrate formation mechanism is different during the different time of a day. The heterogeneous
hydrolysis of $N_2O_5$ was often found to be an important pathway for nighttime nitrate formation.
Here, we evaluated the role of this pathway to nitrate formation in the eastern YRD using the
correlation between particulate nitrate concentration and the production of $N_2O_5$ during nighttime.
Due to the lack of direct observational data of $N_2O_5$ in this study, we used the value of square of
$NO_2$ multiplied by $O_3$ ($[NO_2]^2 \times O_3$) to indicate the $N_2O_5$ level (Liu et al., 2020). Figure 6 shows the
nitrate concentration as a function of $[NO_2]^2 \times O_3$ during the nighttime in winter. The particulate
nitrate concentration showed a strong positive correlation with $[NO_2]^2 \times O_3$. In particular in 2018, as
the value of $[NO_2]^2 \times O_3$ increased to 15000, the nitrate concentration increased from 5-10 μg m$^{-3}$ to
25-30 μg m$^{-3}$, suggesting that the heterogeneous hydrolysis of $N_2O_5$ was an important pathway for
wintertime nitrate formation in the eastern YRD. Notably, there are some data points with low values
of $[NO_2]^2 \times O_3$ but high nitrate concentrations. This might be partly due to their relatively high
aerosol pH (> 3), which could promote the gas-to-particle partitioning of $HNO_3$.



To evaluate the role of the gas-phase OH + $NO_2$ process in nitrate formation during the daytime, we
use the $O_x$ to indicate the atmospheric oxidation capacity due to the lack of direct observational data
of OH radicals. Figure 7 shows the particulate nitrate concentration as a function of $O_x$ during the
daytime. Notably, as the $O_x$ concentration increased, the nitrate concentration also increased
significantly. However, the increase in ALWC seemed to have a relatively small impact on the
nitrate concentration during the daytime, indicating that the reaction of $NO_2$ with OH radicals to
form $HNO_3$ (rather than the gas-to-particle partitioning) was a rate-limiting step in daytime nitrate
formation. We also note that there are some data points with low $O_x$ values but high ALWC and
nitrate concentrations (Figure 7c). This phenomenon might be owing to a certain degree of
heterogeneous process in the haze-foggy days, when the photochemical reactions were relatively
weaker. Overall, the high atmospheric oxidation capacity made the gas-phase OH + $NO_2$ reaction
an important pathway for nitrate formation during the daytime in the eastern YRD.

**3.4 Model constraints on the nitrate formation mechanism**
To quantify the contribution of different formation mechanisms to wintertime nitrate formation in
the eastern YRD, we used an observation-constrained model (F0AM v3.1) updated with the
heterogeneous chemistry of $N_2O_5$ and $NO_2$ (see Section 2.3 for details) to simulate the formation
rate of $HNO_3$ from different pathways during the observation period. During the winter of 2019, six
haze pollution episodes ($PM_{2.5}$ > 75 μg m$^{-3}$) occurred at both sites (there was an additional episode
during the outbreak of COVID-19 epidemic, which was discussed separately in Section 3.5). We
conducted simulations for all the six pollution episodes and took two representative ones at the
Pudong site for the detailed analysis. Considering the large uncertainties in ALWC estimation and
aerosol surface area/volume correction at high RH levels (> 95%), which could significantly affect
the simulation results, we excluded the simulated data above 95% RH from the further analysis.
Figure 8 shows the time series of various particulate (measured) and gaseous (measured and
simulated) air pollutants, as well as the formation rate of $HNO_3$ (simulated) from different pathways
during these two episodes (The case studies of the same episodes at the Qingpu site are given in
Section S4 and Figure S5).

In episode 1 (Figure 8a), the nitrate concentration increased rapidly from 15.2 μg m$^{-3}$ at 22:00 on
December to 39.0 μg m$^{-3}$ at 10:00 on 30 December, with an average growth rate of 2.0 μg m$^{-3}$ h$^{-1}$
. The simulated $NO_2$ concentration was in good agreement with the observation, expect for a short
period around the midnight of 30 December, during which the NO emissions led to an over-
prediction of the $NO_2$ level. During the high nitrate periods, the nighttime $N_2O_5$ concentration could
reach 0.5-1 ppb and contributed noticeably to $HNO_3$ formation via the heterogeneous hydrolysis.
However, the high daytime OH concentration (up to $2.5 \times 10^6$ molecules cm$^{-3}$) facilitated a relatively
more rapid nitrate formation from the gas-phase OH + $NO_2$ pathway. The average production rate
of $HNO_3$ from the gas-phase OH + $NO_2$ reaction during the daytime was 2.9 μg m$^{-3}$ h$^{-1}$, which was
twice the average production rate of $HNO_3$ from the heterogeneous hydrolysis of $N_2O_5$ during the
nighttime.




We note that the overestimation of $NO_2$ during the night of 30 December (case 1) could lead to an
overestimation of nighttime HONO, but it did not significantly affect the overall production rate of
HONO and thereby OH radicals in this case, which was dominated by the daytime heterogeneous
photochemical processes (see Figure S7, HONO production rate in the base scenario). In addition,
as the $O_3$ concentration in the model was constrained by the measured value, which was very low
($< 5$ ppb) during this time, the overestimation of $NO_2$ would also not significantly affect the
prediction of $N_2O_5$. As a result, the over-prediction of $NO_2$ would not have a large influence on the
major formation pathways of nitrate.

There were two cases in the episode 2 (Figure 8b). In case 2, the concentration of nitrate increased
from 26.8 μg m$^{-3}$ at 05:00 to 46.0 μg m$^{-3}$ at 13:00 on 12 January, 2020, with an average growth rate
of 2.4 μg m$^{-3}$ h$^{-1}$. Then, the nitrate concentration achieved a fast growth from 40.2 to 70.5 μg m$^{-3}$
within only six hours during the night of 12 January, with an average rate of 5.1 μg m$^{-3}$ h$^{-1}$. During
the nitrate increasing period, the maximum OH concentration was ~ $1.0 \times 10^6$ molecules cm$^{-3}$. As a
result, the gas-phase OH+$NO_2$ reaction led to a slow increase of nitrate concentration in the daytime
of 12 January. During the nighttime, the $N_2O_5$ concentration quickly increased to 0.83 ppb. The high
$N_2O_5$ level, in combination with the high ALWC, made the heterogeneous hydrolysis of $N_2O_5$ a
more important pathway for nitrate formation. The simulated average production rate of $HNO_3$ from
the heterogeneous hydrolysis of $N_2O_5$ during this case was 4.0 μg m$^{-3}$ h$^{-1}$, which was 3.6 times that
of the formation rate from the gas-phase OH + $NO_2$ reaction (1.1 μg m$^{-3}$ h$^{-1}$). In case 3, the nitrate
concentration increased from 22.5 μg m$^{-3}$ at 0:00 to 53.8 μg m$^{-3}$ at 11:00 on 14 January, with an
average growth rate of 2.8 μg m$^{-3}$ h$^{-1}$. The $N_2O_5$ concentration was at a high level (~ 1 ppb) during
the nighttime and its hydrolysis contributed significantly to nitrate formation at the beginning of the
nitrate-increasing period. In the morning of 14 January, the OH concentration rapidly increased to
$1.3 \times 10^6$ molecules cm$^{-3}$, resulting in considerable nitrate formation from the gas-phase process.
The average production rates of $HNO_3$ from the heterogeneous and gas-phase processes in this case
were 3.9 and 2.4 μg m$^{-3}$ h$^{-1}$, respectively, suggesting that both processes were important nitrate
formation pathways.

As mentioned above, there were six haze pollution episodes during the observation period. At the
Qingpu site, the heterogeneous hydrolysis of $N_2O_5$ was the major formation pathway (65-80%) of
nitrate aerosol for four episodes, while the gas-phase OH + $NO_2$ reaction had a major contribution
(54-60%) for the other two episodes. At the Pudong site, the heterogeneous process also contributed
dominantly (67-89%) to nitrate formation during four episodes, and for the other two episodes, the
contributions of the heterogeneous and gas-phase processes were comparable (51-53% vs. 45-47%).
Figure S6 shows the average diurnal variation of the production rates of $HNO_3$ from different
pathways during the observation period in 2019. The gas-phase process produced $HNO_3$ mainly
from 7:00 to 16:00, while the $HNO_3$ production from the heterogeneous process occurred mainly
from 17:00 to 6:00. The average production rates of $HNO_3$ from the heterogeneous and gas-phase



processes are given in Figure 9. At the Qingpu site, the average production rate of $HNO_3$ from the
two processes was 3.79 μg m$^{-3}$ h$^{-1}$ for the heterogeneous process during the nighttime (14 hours) vs.
2.94 μg m$^{-3}$ h$^{-1}$ for the gas-phase reaction during the daytime (10 hours). The production rate from
other processes such as $NO_2$ hydrolysis and $NO_3$ radical oxidation of VOCs was only 0.08 μg m$^{-3}$
h$^{-1}$. Therefore, the heterogeneous and gas-phase processes contributed to 63% and 35% of nitrate
formation at this site, respectively. At the Pudong site, the average formation rate of $HNO_3$ from the
hydrolysis of $N_2O_5$ was 3.83 μg m$^{-3}$ h$^{-1}$, significantly higher than that from the gas-phase reaction
(2.27 μg m$^{-3}$ h$^{-1}$). As a result, the contributions of heterogeneous and gas-phase processes to nitrate
formation were 69% and 29%, respectively.

It should be noted that significant uncertainties remain in the key parameters of the heterogeneous
HONO formation pathways in the model, which could affect the prediction of the OH level and
thereby gas-phase formation of $HNO_3$. However, sensitive analyses for various parameters show
that the current parameterization of these heterogeneous reactions in the model (see Table 1) allows
for robust quantitative constraints on the relative contributions of the gas-phase and heterogeneous
processes to nitrate formation during haze pollution episodes (see Section S5 and Figure S7 for more
details).

As discussed in Section 3.2, the gas-to-particle partitioning of $HNO_3$ was rather efficient, with the
value of $\varepsilon HNO_3$ larger than 0.9 for 90% of the time during the haze pollution periods. Therefore,
the overall formation rate of particulate nitrate would be determined by the production rate of $HNO_3$
from the heterogeneous hydrolysis of $N_2O_5$ and gas-phase OH + $NO_2$ reaction. To identify the key
chemical factors that controlled the production rates of $HNO_3$ from these two major reaction
pathways, the relationships between the $HNO_3$ production rate and concentrations of $NO_2$ and
oxidants (i.e., $O_3$ or OH radicals) are examined and plotted in Figure 10.

As shown in Figure 10a, the slopes of the $HNO_3$ production rate from the heterogeneous process vs.
$NO_2$ during the nighttime were different under different $O_3$ concentrations. When $O_3$ concentrations
were higher than 10 ppb, the increase in $NO_2$ led to a significant increase in $HNO_3$ production, with
the production rate exceeding 5 μg m$^{-3}$ h$^{-1}$ when the $NO_2$ was higher than 30 ppb. However, when
the $O_3$ level was low (< 10 ppb), the heterogeneous process was relatively slow, even with $NO_2$
concentration exceeding 60 ppb. These results suggest that the atmospheric oxidation capacity (or
the availability of $O_3$), which affected the production of $N_2O_5$, played a vital role in controlling the
nitrate formation rate from the heterogeneous process. Furthermore, the reactive uptake of $N_2O_5$ by
aerosols was found to be very efficient (see Figure S8) so that it was not the rate-limiting step of the
heterogeneous nitrate formation during the haze pollution periods. Similarly, the slope of the $HNO_3$
production rate from the gas-phase process vs. $NO_2$ during the daytime also varied dramatically
under different OH radical concentrations (Figure 10b). As the OH radical concentration was higher
than $7 \times 10^5$ molecules cm$^{-3}$, this rate increased markedly with the increase in $NO_2$. This
phenomenon proved again that the atmospheric oxidation capacity played a driving role in the



production of HNO$_3$ from the gas-phase process.

The results in Figure 10 also suggest that solely reducing the NO$_x$ emissions might result in an
increase of O$_3$ and OH concentrations, which could enhance the oxidation of NO$_x$ and thereby offset
the effect of NO$_x$ emission reductions on HNO$_3$ production. Therefore, a synergistic control of
atmospheric oxidant and NO$_x$ emissions would be of great importance for mitigating wintertime
particulate nitrate pollution in the eastern YRD.

**3.5 Nitrate aerosol formation during the COVID-19**
The city lockdowns during the COVID-19 epidemic resulted in substantial emission reductions from
vehicular and industrial sources, which provided an opportunity to investigate the response of
secondary aerosols to primary emission reductions. Here, we selected the 23 January, 2019 as a
demarcation point (since then many cities in China started to implement lockdown measures) and
analyzed the characteristics of particulate nitrate pollution before and during the COVID-19
epidemic.

Figure 11 shows the concentrations of major gaseous and particulate air pollutants, NOR, and sulfur
oxidation ratio (SOR) in the eastern YRD before (1-22 January, 2020) and during (23 January-12
February, 2020) the COVID-19 epidemic. At the Pudong site (Figure 11 a, b, c), the average NO$_x$
concentration decreased by 57% due to marked reductions in vehicular emissions during the
epidemic. In contrast, the SO$_2$ concentration only had a small decrease (16%) during the epidemic,
since it mainly comes from coal-combustion sources and is less affected by vehicular emissions.
However, the O$_3$ concentration increased by 66% during the epidemic. This is mainly due to the
significant reduction in NO$_x$ emissions, though the changes in meteorological conditions could also
contribute (Zhao et al., 2020b). Accordingly, the model simulations show that the atmospheric OH
concentration (median) increased by 14% during the epidemic, though the average value only
increased slightly. The increase in O$_3$ and OH concentrations could significantly promote the
oxidation of NO$_x$ to nitrate and SO$_2$ to sulfate through both gas-phase and heterogeneous processes.
As shown in Figure 11c, the average values of NOR and SOR increased from 0.15 and 0.46 before
the epidemic to 0.21 and 0.50 during the epidemic, respectively. The enhanced oxidation of NO$_x$
and SO$_2$ would weaken the response of particulate nitrate and sulfate to the emission reductions. As
can be seen in Figure 11b and c, the simulated HNO$_3$ production rate and measured particulate nitrate
concentration dropped by 42% and 40% during the epidemic, respectively, which were both
significantly smaller than the decrease in NO$_x$ concentration (57%), while the particulate sulfate
concentration only decreased by 2%, also substantially smaller than the reduction in SO$_2$
concentration (16%).

Similarly, at the Qingpu site, the NO$_x$ concentration decreased by 58% during the epidemic, while
the concentrations of O$_3$ and OH radicals (median) increased by 90% and 17%, respectively. The
significantly enhanced atmospheric oxidation capacity made the simulated HNO$_3$ production rate



only decrease by 17% during the epidemic. However, the measured particulate nitrate concentration
at this site decreased by 60%, comparable to the decrease in $NO_x$ concentration. The inconsistency
between the decrease in measured nitrate concentration and simulated $HNO_3$ production rate at the
Qingpu site was different from the situation observed at the Pudong site, which is likely due to the
fact that the Qingpu site was more easily to be influenced by the regional transport. We note that the
average wind speed at the Qingpu site (1.8 m s$^{-1}$) was higher than that at the Pudong site (1.1 m s$^{-1}$
). Besides, the haze pollution was more serious at the Qingpu site than at the Pudong site before
the epidemic: both $PM_{2.5}$ and nitrate concentrations were significantly higher at the Qingpu site (see
Figure 11). Therefore, the marked emission reductions on a regional scale during the epidemic
would decrease both the local formation and transport of particulate nitrate from the upwind regions,
resulting in a more pronounced reduction in observed nitrate concentration at the Qingpu site. In
addition, as air plume influenced by the regional transport was more aged, the NOR and SOR values
before the epidemic were even higher than those during the epidemic.
The results at the Pudong site clearly show that the enhanced atmospheric oxidation capacity during
the COVID-19 epidemic promoted the formation of secondary aerosols and offset the effects of
primary emission reductions in the eastern YRD. Such a phenomenon has also been observed in
many other regions in China during the COVID-19 lockdown (Le et al., 2020; Zheng et al., 2020;
Huang et al., 2021; Liu et al., 2021; Tian et al., 2021; Zhong et al., 2021). These results suggest an
important role of atmospheric oxidation capacity in regulating secondary aerosol formation. They
also highlight the importance of the synergetic regulation of atmospheric oxidants and other air
pollutants in the mitigation of particulate pollution in China. However, the Qingpu site also provided
us a special case that in severely polluted regions with a stronger influence from the regional
transport, the offset effects of enhanced atmospheric oxidation capacity on emission reductions
could be more complicated and less significant.
**4. Conclusions**
In this study, the chemical mechanisms and key controlling factors of wintertime nitrate formation
in the eastern YRD of China were investigated using a combination of online field observations and
detailed model simulations. During the observation period (Winter 2018 and 2019), the haze
pollution events ($PM_{2.5} > 75$ μg m$^{-3}$) occurred frequently in this region. The mass fraction of nitrate
in $PM_{2.5}$ increased dramatically with $PM_{2.5}$ concentration and exceeded 30% throughout the
pollution periods. The measured nitrate concentration was well correlated with $[NO_2]^2 \times [O_3]$ (an
indicator of $N_2O_5$) at night and the level of $O_x$ (an indicator of atmospheric oxidation capacity)
during the daytime, indicating that both the heterogeneous hydrolysis of $N_2O_5$ and gas-phase OH +
$NO_2$ process played important roles in wintertime nitrate formation in the eastern YRD.
Observation-constrained model simulations further show that the average production rates of $HNO_3$
from the heterogeneous hydrolysis of $N_2O_5$ during the nighttime and gas-phase OH + $NO_2$ reaction
during the daytime were 3.81 μg m$^{-3}$ h$^{-1}$ and 2.61 μg m$^{-3}$ h$^{-1}$, respectively, during the haze pollution
periods; these two pathways accounted for 66% and 32% of wintertime nitrate formation in the



eastern YRD, respectively.

The ALWC significantly promoted the formation of nitrate by facilitating the hydrolysis of $N_2O_5$
and the gas-to-particle partitioning of $HNO_3$. However, the promoting effect of ALWC on nitrate
formation varied with aerosol pH due to its significant influence on the gas-to-particle partitioning
of $HNO_3$. During the pollution periods, the gas-to-particle partitioning of $HNO_3$ was very efficient,
with the partitioning coefficients, $\varepsilon HNO_3$, larger than 0.9 for 90% of the time. Therefore, the overall
formation processes of wintertime particulate nitrate were not limited by the gas-to-particle
partitioning of $HNO_3$ but rather by its production from both heterogeneous and gas-phase processes.
Further analyses of the response of $HNO_3$ formation to the variation in the concentrations of $NO_2$,
$O_3$, and OH radicals suggests that the atmospheric oxidation capacity (i.e., the availability of $O_3$ and
OH radicals) played a key role in controlling the formation of nitrate from both processes.

During the COVID-19 lockdown (January-February 2020), the enhanced atmospheric oxidation
capacity promoted the oxidation of $NO_x$ to nitrate and weaken the effects of primary emission
reductions on particulate pollution in typical urban areas in the eastern YRD, though such an offset
effect was less significant in regions with a stronger influence from the regional transport. This
phenomenon again suggests that the atmospheric oxidation capacity played an important role in
driving the formation of secondary aerosols, and highlights the importance of the synergetic
regulation of atmospheric oxidation capacity and other air pollutants in the mitigation of particulate
pollution in eastern China.

*Data availability.* The data presented in this work are available upon request from the corresponding
authors.

*Author contributions.* YZ designed the study, JH, QZ, QF, and YD performed field measurements,
JYS conducted ISORROPIA-II model calculation, JA and CH provided the $NO_x$ emission inventory,
and YZ and HZ analyzed the data, conducted model simulations, and wrote the paper. All other
authors contributed to discussion and writing.

*Competing interests.* The authors declare no conflict of interest.

*Acknowledgments.* This work was supported by the National Natural Science Foundation
of China (Grant # 22022607) and the Science and Technology Commission of Shanghai
Municipality (Grant # 19DZ1205004). Yue Zhao acknowledges the Program for Professor
of Special Appointment (Eastern Scholar) at Shanghai Institutions of Higher Learning.

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





Table 1 Parameterization of the formation and removal pathways of HONO added to the model.

| Mechanism | Parametrization | Max | Min | Ref |
|---|---|---|---|---|
| $NO_2$+aerosol→0.5HONO+0.5HNO$_3$ | $\gamma NO_2 = 2 \times 10^{-6}$ | $1 \times 10^{-5}$ | $4 \times 10^{-7}$ | a-d |
| $NO_2$+ground→HONO | $\gamma NO_2 = 2 \times 10^{-6}$ | $1 \times 10^{-5}$ | $4 \times 10^{-7}$ | a-d |
| $NO_2$+aerosol+hv→HONO | $\gamma NO_2 = 2 \times 10^{-5} \times jNO_2/jNO_2noon^*$ | $1 \times 10^{-4}$ | $4 \times 10^{-6}$ | b, e-g |
| $NO_2$+ground+hv→HONO | $\gamma NO_2 = 2 \times 10^{-5} \times jNO_2/jNO_2noon^*$ | $1 \times 10^{-4}$ | $4 \times 10^{-6}$ | b, e-g |
| pNO$_3$+hv→HONO | $jNO_3 = jHNO_3 \times 30$ | 100 | 1 | h, i |
| Vehicular emission | HONO/NO$_x$=0.8% | 0.18% | 1.6% | j-l |
| $NO_2$+SO$_2$+aerosol→HONO+SO$_4^{2-}$ | $k_{aq} = 1.4 \times 10^5\,M^{-1}\,s^{-1}\ (pH < 5);$ $2 \times 10^6\,M^{-1}\,s^{-1}\ (pH > 6)$ | | | m, n |
| HONO deposition | $k_{dep} = \exp^{(23920/T-91.5)}/PBL$ | | | a |

*The value of jNO$_2$noon used in the model was 0.005 s$^{-1}$; References: [a]Xue et al. (2020); [b]Liu et al.
(2019); [c]Wong et al. (2011); [d]Kleffmann et al. (1998); [e]Wong et al. (2013); [f]Zare et al. (2018); [g]Han
et al. (2016); [h]Romer et al. (2018); [i]Ye et al. (2016); [j]Kurtenbach et al. (2001); [k]Liu et al. (2017),
[l]Trinh et al. (2017); [m]Lee and Schwartz (1983); [n]Wang et al. (2020a).



Table 2 Concentrations (average ± standard deviation) of PM$_{2.5}$, particulate nitrate, NO$_x$, and O$_3$, as
well as temperature and RH at Qingpu and Pudong sites in the winter of 2018 and 2019.

| | Sites | | | |
|---|---|---|---|---|
| | Qingpu-2018 | Pudong-2018 | Qingpu-2019 | Pudong-2019 |
| PM$_{2.5}$ (µg m$^{-3}$) | 50.0 ± 34.8 | 40.9 ± 32.5 | 58.6 ± 37.2 | 49.5 ± 35.3 |
| NO$_3^-$ (µg m$^{-3}$) | 14.9 ± 12.8 | 11.9 ± 12.2 | 17.0 ± 14.8 | 13.2 ± 12.0 |
| NO$_x$ (ppb) | 29.6 ± 31.1 | 27.5 ± 24.4 | 35.1 ± 33.1 | 26.9 ± 21.3 |
| O$_3$ (ppb) | 19.1 ± 12.7 | 18.8 ± 10.4 | 21.7 ± 14.3 | 22.3 ± 12.0 |
| Temperature (°C) | 6.6 ± 4.4 | 7.3 ± 4.2 | 7.5 ± 4.2 | 8.2 ± 3.8 |
| RH (%) | 80 ± 17 | 78 ± 18 | 80 ± 17 | 79 ± 20 |






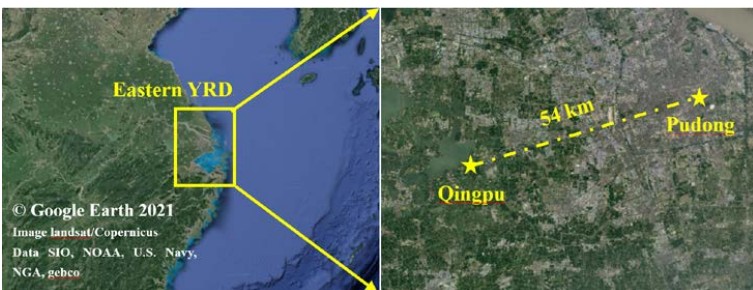


Figure 1 Map of the eastern YRD region and the two observation sites, i.e., Qingpu (suburban and
regional) and Pudong (urban).






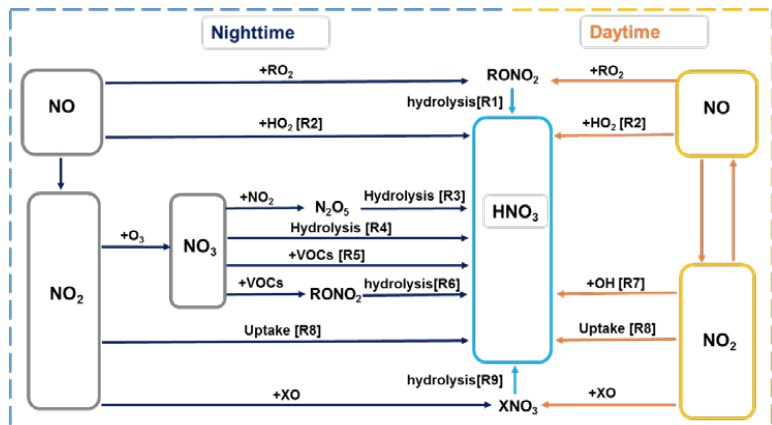

Figure 2 Simplified HNO$_3$ formation mechanisms in the troposphere. X represents Cl, Br, and I.



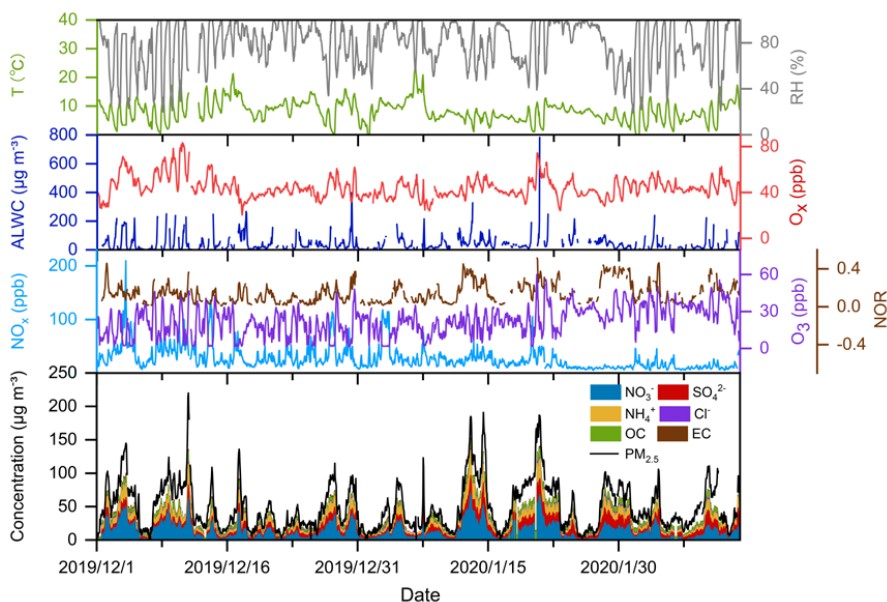


Figure 3 Time series of temperature, relative huidity (RH), aerosol liquid water content (ALWC),
$NO_x$, $O_3$, $O_x$, nitrogen oxidation ratio (NOR), as well as $PM_{2.5}$ and major particulate compositions
at the Pudong site in the winter of 2019.





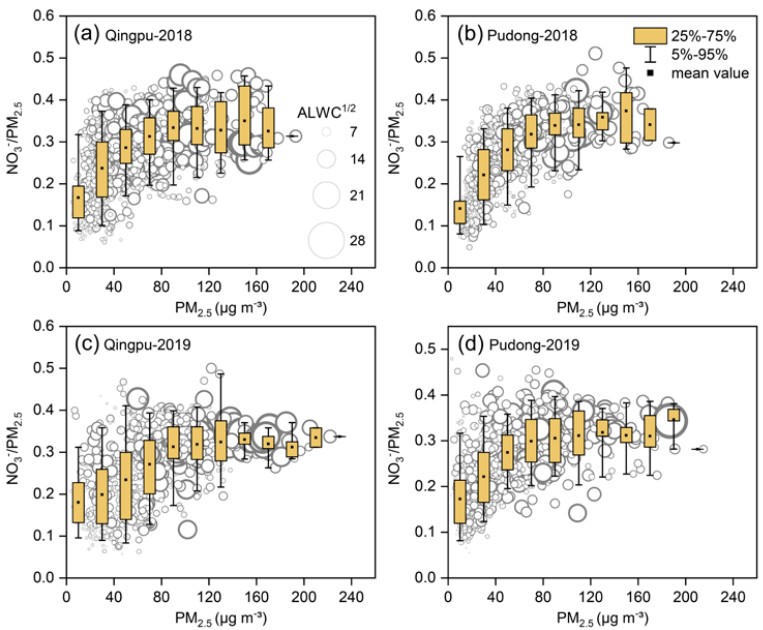


Figure 4 Mass ratio of nitrate to PM$_{2.5}$ as a function of PM$_{2.5}$ concentration at (a, c) Qingpu and (b,

d) Pudong sites in the winter of 2018 and 2019. The circles represent the measured ratio of NO$_3^-$

/PM$_{2.5}$, and their area is linearly scaled with square root of ALWC.


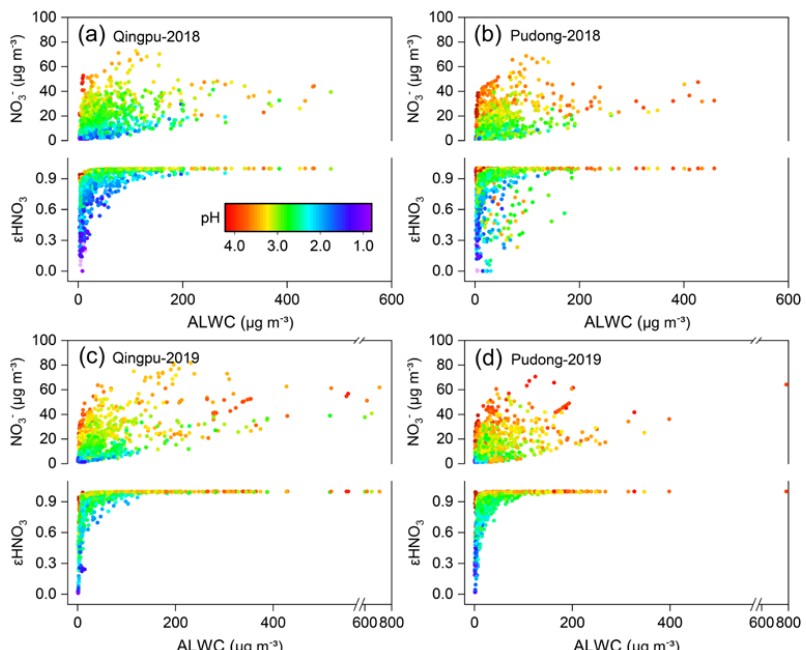


Figure 5 Particulate nitrate concentration and its fraction to total nitrate ($\varepsilon HNO_3$) as a function of
ALWC and aerosol pH at (a, c) Qingpu and (b, d) Pudong sites in the winter of 2018 and 2019. The
circles are colored according to aerosol pH.



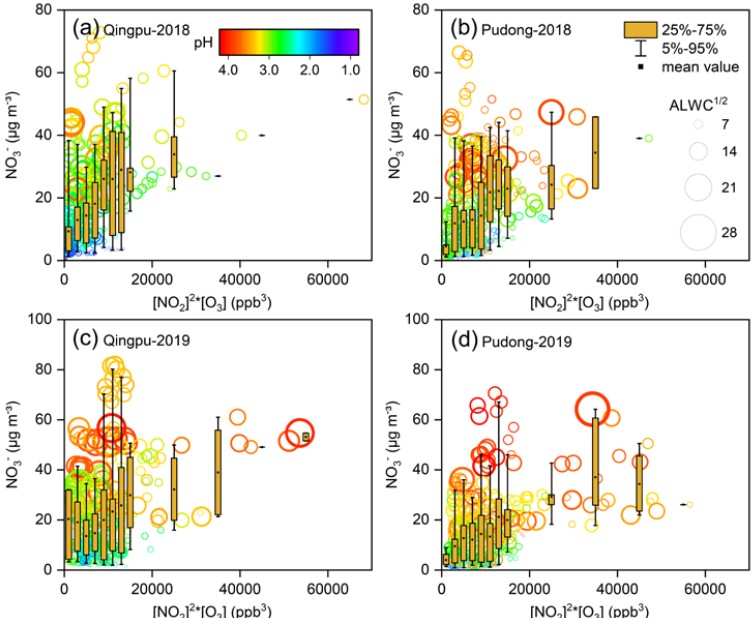


Figure 6 Particulate nitrate concentration as a function of $[NO_2]^2 \times [O_3]$ during the nighttime at (a, c)
Qingpu and (b, d) Pudong sites in 2018 and 2019. The circles are colored according to aerosol pH
and their size is linearly scaled with square root of ALWC.



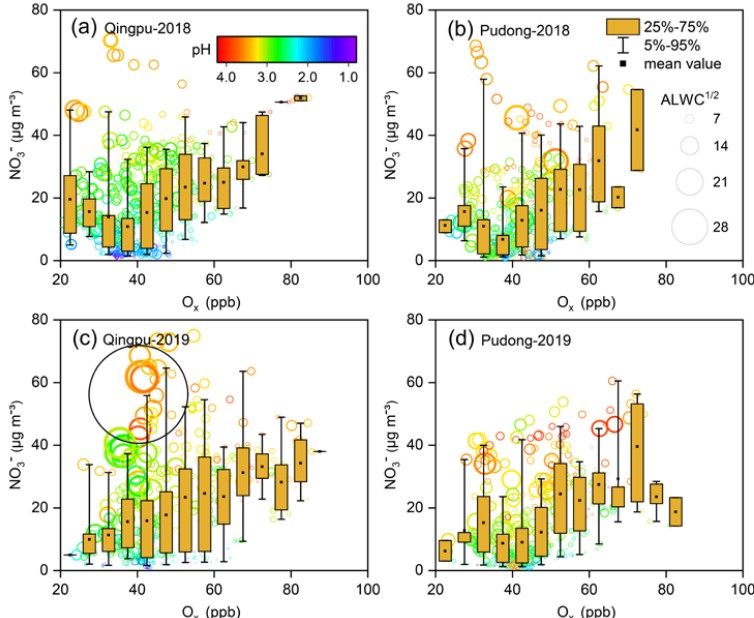


Figure 7 Particulate nitrate concentration as a function of $O_x$ during the daytime at (a, c) Qingpu
and (b, d) Pudong sites in 2018 and 2019. The circles are colored according to aerosol pH and their
size is linearly scaled with square root of ALWC. The data points inside the black circle in (c)
correspond to low $O_x$ levels but high ALWC and nitrate concentrations.






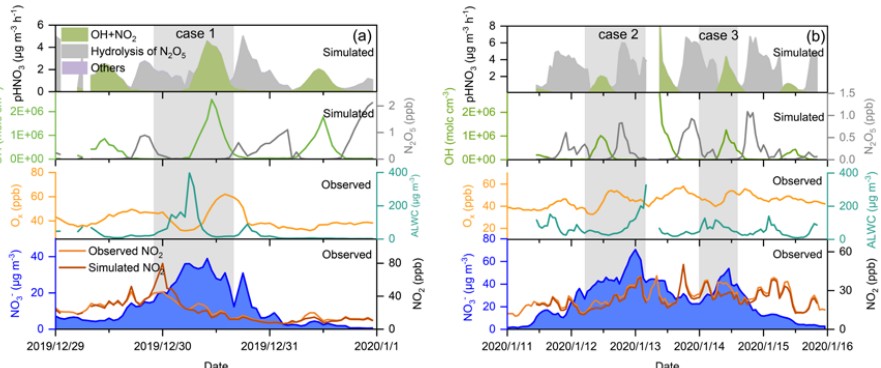

Figure 8 Time series of particulate nitrate, $NO_2$, $O_x$, ALWC, OH, $N_2O_5$, as well as the formation rate
of $HNO_3$ from different processes during the two selected case during the pollution episodes at the
Pudong site in 2019. The simulated data with RH > 95% were not included in the figure (see main
text).



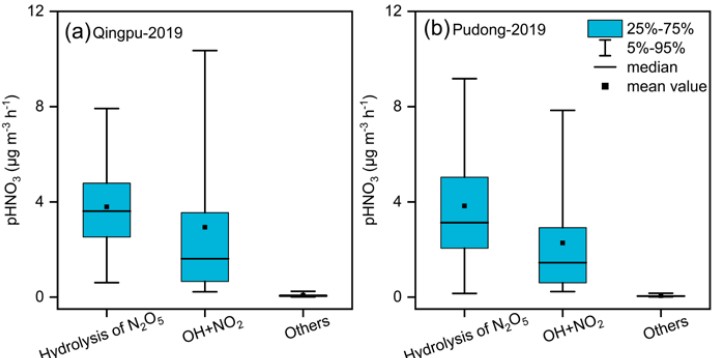


Figure 9 Simulated average formation rates of HNO$_3$ at (a) Qingpu and (b) Pudong sites during the
haze pollution periods in 2019

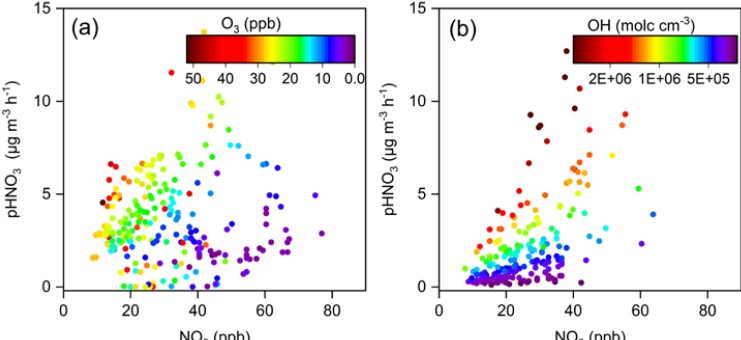

Figure 10 Production rates of HNO$_3$ from the (a) heterogeneous and (b) gas-phase processes as a function of NO$_2$ concentration at the Pudong site during the nighttime and daytime, respectively. The circles are colored according to the O$_3$ concentration in (a) and OH radical concentration in (b).



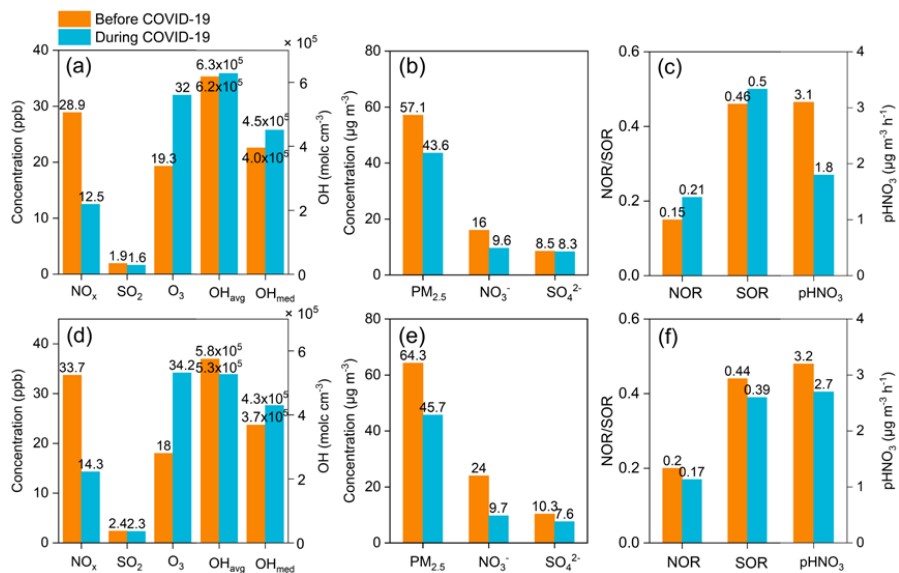

942

Figure 11 Average concentrations of $NO_x$, $SO_2$, $O_3$, OH radicals, $PM_{2.5}$, nitrate, sulfate, as well as

the nitrogen and sulfur oxidation ratio (NOR and SOR) at (a-c) Pudong and (d-f) Qingpu sites before

(1-22 January, 2020) and during (23 January-12 February, 2020) the COVID-19 epidemic.