# Peer review of "High atmospheric oxidation capacity drives wintertime nitrate pollution in the eastern Yangtze River Delta of China"

_Atmospheric Chemistry and Physics, 2021_

## Author Comment (AC2)

**Response to Reviewer #2**

We are grateful to the reviewer for the thoughtful comments on the manuscript. Our point-to-point responses to each comment are as follows (reviewer's comments are in black font and our responses are in blue font).

**Comments:**

Zang et al., present a comprehensive study to identify the major nitrate formation pathways and their key controlling factors during the winter haze pollution period in the eastern YRD, China using two-year (2018-2019) field observations and detailed observation-constrained model simulations. They find that high atmospheric oxidation capacity is the reason for the winter nitrate aerosol pollution in YRD region in China. And N2O5 uptake contributes 60-70% in urban and suburban sites in polluted days. The analysis of the observation data is sound, I only have some comments to the model simulations.

**Major Issues:**

Line 24-27, The quantification of nitrate formation importance is derived from pollution episodes only. The campaign average result should be much more different. Please clarify it.

Response: Thanks for the reviewer's comment. In this study, we focused on the contribution of different processes to nitrate formation during the haze pollution episodes. To be more precise, we have revised this description as "We find that..., with contribution percentages of 69% and 29% in urban areas and 63% and 35% in suburban areas during the haze pollution episodes, respectively." (changes underlined).

The model includes the dry deposition of HNO3, it seems that the authors want to simulate the variation the particle nitrate. I am very interesting whether the modelled nitrate comparable with the observation. Is it possible to provide more details about the intercomparison? In addition, when calculating the contribution of nitrate formation, are you just accumulate the nitrate production rate during a certain period from different channel? Are the only represent the formation potential without considering the dry deposition, what is the role of the dry deposition in the model simulation since it cannot influence any result in the paper?

Response: In this study, we simulated the formation rate (i.e., formation potential) of HNO3 from different pathways but not the concentration of particulate nitrate. Accordingly, the contribution of nitrate formation was the accumulation of the HNO3 production rate from different channel over a certain period (e.g., daytime or nighttime). In the manuscript, we have compared the increasing rates of particulate nitrate with the formation rates of HNO3 for several typical episodes and found that the two rates were comparable. The dry deposition did not influence the formation potential of HNO3, so we have removed its calculation in Section 2.3 of the main text.

The heterogeneous chemistry is well considered in the model simulation, such as the  $N_2O_5$  and  $NO_2$  uptake mechanism, but limited by the observation, the importance of these reactions cannot be confirmed, If the field measurement of  $N_2O_5$  or ClNO2 are available, the result would be more insightful with smaller uncertainties. Here, I suggest the author provide more information about the parameterized  $N_2O_5$  uptake and ClNO2 yield in the main text or SI, which could help people to connect the further observation studies that quantifying  $N_2O_5$  uptake coefficient and/or ClNO2 yield.

Response: We certainly agree that simultaneous measurements of  $N_2O_5$  and  $CINO_2$  would provide strong constraints on the nitrate formation chemistry, but unfortunately such measurements are not available in this study. Instead, we carefully parameterized the heterogeneous nitrate formation pathways based on recent advances on the reaction kinetics and well-measured aerosol data. As mentioned in our replies to the previous comment, the modelled HNO3 production rates were comparable to the measured increasing rates of particulate nitrate during several pollution episodes, indicating our model results are reliable.

According to the reviewer' suggestion, we have added more information about the parameterized N2O5 uptake and ClNO2 yield in Section 2.3 of the revised manuscript (changes underlined).

"For the heterogeneous hydrolysis of  $N_2O_5$ , the  $N_2O_5$  molecules accommodated on aqueous aerosols can undergo reversible hydrolysis to form  $NO_3^-$  and  $H_2ONO_2^+$  (R1), followed by the reaction of  $H_2ONO_2^+$ with  $H_2O$  or Cl- to form HNO3 and ClNO2 (R2 and R3) (Finlayson-Pitts et al., 1989; Schweitzer et al., 1998; Thornton and Abbatt, 2005):

$$N_2O_5(aq) + H_2O(l) \xrightarrow{k_{1f}} H_2ONO_2^+ (aq) + NO_3^- (aq)$$
 (R1)

$$H_2ONO_2^+ (aq) + H_2O(l) \xrightarrow{k_2} H_3O^+(aq) + HNO_3(aq)$$
(R2)

$$H_2ONO_2^+ (aq) + Cl^-(aq) \xrightarrow{\kappa_3} ClNO_2(g) + H_2O(l)$$
(R3)

•••

$$\gamma N_2 O_5 = \frac{4}{c} \frac{V_a}{S_a} K_H \times k_{1f} \times (1 - \frac{1}{\left(\frac{k_2}{k_{1b}} \times \frac{[H_2 O]}{[NO_3]}\right) + 1 + \left(\frac{k_3}{k_{1b}} \times \frac{[Cl^{-}]}{[NO_3]}\right)}) \quad (3)$$

where  $V_a$  is the measured aerosol volume concentration;  $K_H$  is the Henry's law coefficient of N2O5, with a value of 51 M atm-1 (Bertram and Thornton, 2009);  $k_{lf}$  is the second-order reaction rate constant of N2O5 with water, which was calculated using a linear function with [H2O], as  $3.0 \times 10^4 \times [\text{H}_2\text{O}]$  (Yu et al., 2020a);  $\frac{k_2}{k_{1b}}$  and  $\frac{k_3}{k_{1b}}$  are the relative rates of reactions of H2ONO+2(aq) with H2O or Cl- (R2 and R3) versus that with NO-3 (the reverse reaction of R1), with values determined to be 0.033 and 3.4, respectively (Yu et al., 2020a); and [H2O], [NO+3], and [Cl-] are the molarity of water, nitrate, and chloride in aerosol, respectively.

•••

$$\Phi_{HNO3} = 1 - 1/(1 + \frac{[H_2 0]}{\frac{k_3}{k_2} \times [Cl^-]})$$
(4)

where  $\frac{k_3}{k_2}$  is the ratio of reaction rates of R3 versus R2, which has been determined to be 105 (Bertram and Thornton, 2009; Yu et al., 2020)."

Monoterpene is very reactive to  $NO_3$  radical, and we notice that monoterpene was not included in the model simulation, although the monoterpenes concentration may be low during the winter due to low temperature, but it maybe still have large contribution to the  $NO_3$  loss and affect the budget, I encourage the authors do some sensitivity tests to assess the impacts to  $N_2O_5$  uptake and following nitrate formation.

Response: We appreciate the reviewer's point. We have conducted a sensitivity test for monoterpenes to evaluate their influence on the HNO3 formation. It should be noted that we only have the observation data of monoterpenes obtained using a proton transfer reaction time-of-flight mass spectrometry (PTR-ToF-MS, Vocus, Tofwerk) at an urban site in Shanghai in early November, 2019. We selected the data on 9 November as the ambient temperature (average: 13.3 °C) that strongly affects monoterpene emissions (Guenther et al., 2012), was relatively low on this day, close to the temperature in the winter. The wind speed (average: 0.76 m s-1) was also low on this day, which limits the transport and dilution of monoterpene emissions. The monoterpene concentration on this day ranges from 0.009 ppb to 0.070 ppb, with an average of 0.038 ppb. The sensitivity analysis shows that when the monoterpene chemistry was considered, the N2O5 concentration and HNO3 production rate from N2O5 hydrolysis (pHNO3(N2O5)) both had a decrease, especially during the nighttime with high N2O5 concentration (Figure S9a, b). However, such decrease was relatively small; the average N2O5 concentration and pHNO3(N2O5) decreased by 23% and 12% during the nighttime, respectively. In addition, the contribution of heterogeneous N2O5 hydrolysis to HNO3 formation only decreased by 2.7% (Figure S9c). Notably, the average temperature in the selected winter haze episode was 8.1 °C, which was significantly lower than the temperature on 9 November, so the concentration of monoterpenes should be smaller, as is their impact on the HNO3 formation. To sum up, the low monoterpene emissions had no significant impact on the budget of NO3 radicals and N2O5 as well as the formation of HNO3 during the winter haze pollution episodes in eastern YRD.

In the revised manuscript, we have added the above sensitivity analyses and Figure R1 to the supplement as a new section (Section S6. Potential influences of monoterpenes on HNO3 production). In addition, we have added the following discussion to Section 3.4 of the main text.

"In addition, monoterpenes that are very reactive to NO3 radicals (Atkinson and Arey, 2003) were not included in the model, because their measurements are not available in this study. However, a case study

considering the monoterpene chemistry in the model shows that the low monoterpene emissions during the winter did not significantly affect the budget of  $NO_3$  radical and  $N_2O_5$  and thereby the nighttime HNO3 production (see Section S6 and Figure S9 for more details)."

Figure R1 Sensitivity of  $N_2O_5$  concentration, production rates of HNO3 from  $N_2O_5$  hydrolysis (pHNO3(N2O5)), as well as its contribution to the HNO3 formation (pHNO3(N2O5)/ pHNO3(total)), to the inclusion of monoterpenes in the model simulation. The chosen episode was from 26 to 31 December, 2019. The base case did not consider the effect of monoterpenes.

Line 249-250, why only constrain the sum NO and NO2, if the NO and NO2 not constrained separately but only the sum, I guess the modeled nocturnal NO always be zero when O3 over ppb. While in fact NO spikes by local emission always observed in urban regions during the nighttime, which would lead to a bias of nitrate formation from N2O5 uptake (possibly an overestimation).

Response: Thanks for the reviewer's comment. We have tried to constrain NO and NO2 separately in the model, but when we did this, the simulated nighttime concentrations of NO3 radical and N2O5 were extraordinarily low during the whole observation period, owing to the titration of NO3 by NO. In addition, high N2O5 peaks were simulated during the daytime likely due to the high O3 concentrations in the model, which is unreasonable.

Therefore, we constrained the sum NO and NO2 but let their specific ratios be simulated by the model. As shown in Figure 8 in the main text and Figure S5 in the supplement, the simulated NO2 concentration was generally in good agreement with the observation, which would also be the case for NO given that the sum of NO and NO2 was constrained by observation. The NO spikes did exist during the nighttime in some episodes, which could lead to an overestimation of NO2. However, as discussed in the manuscript, as the O3 concentration in the model was constrained by the observation, which was very low (below 5 ppb) during the NO spikes periods, the overestimation of NO2 did not significantly affect the prediction of N2O5.

Figure 8 case 1, the observed  $NO_2$  during daytime and nighttime had a lower and higher biases, are they mean the modelled nitrate during the daytime is lower and nighttime is higher. This phenomenon also happened in case 3.

Response: The overestimation of NO2 during the nighttime was due to the NO spikes. As we explained in the previous comment, the O3 concentration in the model was constrained by the measured value, which was very low, the overestimation of NO2 did not significantly affect the modelled N2O5 and its contribution to HNO3 formation. The bias of modelled NO2 during the daytime is quite small compared to that during the nighttime, it therefore might also have no significant impact on the model results.

Line 81 or change to "and"

Response: We have revised this.

Line 361 weaker change to "weak".

Response: We have revised this.

Line 346 the value 15000 misses the unit, may be pppbv3.

Response: We have added the unit ppb3 for the value.

**References:**

- Atkinson, R., and Arey, J.: Atmospheric degradation of volatile organic compounds, Chem. Rev., 103, 4605-4638, doi, 2003.
- Bertram, T. H., and Thornton, J. A.: Toward a general parameterization of N2O5 reactivity on aqueous particles: the competing effects of particle liquid water, nitrate and chloride, Atmos. Chem. Phys., 9, 8351-8363, doi, 2009.
- Finlayson-Pitts, B. J., Ezell, M. J., and Pitts, J. N.: Formation of chemically active chlorine compounds by reactions of atmospheric NaCl particles with gaseous N2O5 and ClONO2, Nature, 337, 241-244, doi: 10.1038/337241a0, 1989.
- Guenther, A. B., Jiang, X., Heald, C. L., Sakulyanontvittaya, T., Duhl, T., Emmons, L. K., and Wang, X.: The Model of Emissions of Gases and Aerosols from Nature version 2.1 (MEGAN2.1): an extended and updated framework for modeling biogenic emissions, Geoscientific Model Development, 5, 1471-1492, doi, 2012.
- Schweitzer, F., Mirabel, P., and George, C.: Multiphase chemistry of N2O5, ClNO2, and BrNO2, J. Phys. Chem. A, 102, 3942-3952, doi: DOI 10.1021/jp980748s, 1998.
- Thornton, J. A., and Abbatt, J. P. D.: N2O5 reaction on submicron sea salt aerosol: Kinetics, products, and the effect of surface active organics, J. Phys. Chem. A, 109, 10004-10012, doi: 10.1021/jp054183t, 2005.
- Yu, C., Wang, Z., Xia, M., Fu, X., Wang, W., Tham, Y. J., Chen, T., Zheng, P., Li, H., Shan, Y., Wang, X., Xue, L., Zhou, Y., Yue, D., Ou, Y., Gao, J., Lu, K., Brown, S. S., Zhang, Y., and Wang, T.: Heterogeneous N2O5 reactions on atmospheric aerosols at four Chinese sites: improving model representation of uptake parameters, Atmospheric Chemistry and Physics, 20, 4367-4378, doi: 10.5194/acp-20-4367-2020, 2020.

---

## Author Response (AR1)

We are grateful to the reviewer for the thoughtful comments on the manuscript. Our point-to-point responses to each comment are as follows (reviewer's comments are in black font and our responses are in blue font).

Comments:

General comments:

This study investigates the key controlling factors nitrate formation in YRD region during wintertime based on field observation and box model. They found large ALWC significantly promoted the uptake of $N_2O_5$ and gas-to-partition of gaseous $HNO_3$, the partitioning coefficient of which varied with pH values of particles. The model calculation showed that $N_2O_5$ uptake contribute to the major fraction of particulate nitrate formation in this region during the pollution periods. Further analysis on the correlation of nitrate with its precursors indicated the controlling effect on nitrate formation resulted from atmospheric oxidation, which could be the availability of ozone and OH radical. A comparison over various parameters associated with nitrate formation made between the data before and during the epidemics also provided confidence for the results derived above.

Overall, this work provides valuable data on analyzing nitrate formation. It shows the dominant contribution from $N_2O_5$ uptake to nitrate formation in YRD region which might be different from other regions in China, and reinforces the importance of atmospheric oxidation on mitigating secondary pollution. I would recommend publication of this paper in Atmospheric Chemistry and Physics after the following comments are well addressed.

Specific comments:

Line 240~243: A constant dilution rate for model is inappropriate. For example, the dilution should be significantly enhanced during the breakup of nocturnal boundary layer in the morning at sunrise. It therefore could influence the calculated abundance of long lifetime species, like particulate nitrate, and change the relative contribution from different pathways. Suggest the parameterization of dilution rate constant varying with PBL for a more accurate quantification.

Response: We agree that the evolution of planetary boundary layer (PBL) has a significant influence on the dilution process. When the PBL increases, the loss of species by dilution can be estimated by:

$$\frac{d[X]}{dt} = -\frac{\partial PBL(t)}{PBL(t) \times \partial t}(X - X_{FT}) \qquad (1)$$

Where $X_{FT}$ is the concentration of X in the residual layer or free troposphere. However, we did not have the measured data of $X_{FT}$, so it is difficult to parameterize this value in the model. If we set $X_{FT}$ to 0 or other values for simplicity, there might be significant uncertainties in the model results. Therefore, to evaluate the influence of the parameterization of dilution rate constant ($k_{dil}$) on the $HNO_3$ production rate from different pathways, we performed a sensitivity analysis for $k_{dil}$ by varying its value from 0.028 $h^{-1}$ to 0.2 $h^{-1}$ (corresponding to a dilution lifetime of 5 hours to 36 hours), which covers the typical range of $k_{dil}$ used in observation-constrained model simulations in the literature (Romer et al., 2018; McDuffie et al., 2019; Liu et al., 2020).

The results of sensitivity analyses during typical pollution episodes are shown in Figure R1. As the dilution lifetime varied from 5 hours to 36 hours, the average concentrations of $N_2O_5$ and OH radicals changed within -23%/+1% and -21.6%/+10.8%, respectively (Figure R1a, d), compared to the base case (dilution lifetime: 24 hours) during the episode. Accordingly, the $HNO_3$ production rates from the heterogeneous hydrolysis of $N_2O_5$ and gas-phase OH + $NO_2$ reactions changed within -17%/+1.2% and -33%/+12% (Figure R1b, e) and the relative contributions of the two pathways changed within -2.5%/+5.5% and -5%/+2.3% (Figure R1c, f), respectively. The relatively small changes in the rates and relative contributions of the two $HNO_3$ production pathways upon variations in $k_{dil}$ from 0.028 $h^{-1}$ to 0.2 $h^{-1}$ suggest that the simplified parameterization of the dilution process using a constant $k_{dil}$ would not result in significant uncertainty in the model results.

In the revised manuscript, we have added the following sentence to Section 2.3 of the main text.

"Considering the uncertainties in the parameterization of dilution process using a constant rate constant, we also conducted a sensitivity test for $k_{dil}$ with its value ranging from 0.028 $h^{-1}$ to 0.2 $h^{-1}$, which covers the typical values used in box model simulations to evaluate its influence on the model results."

In addition, we have added above sensitivity analysis results and Figure R1 to the supplement and rephrased discussions on the results of sensitivity tests in Section 3.4 of the main text (changes underlined).

"Significant uncertainties remain in the key parameters of the heterogeneous HONO formation pathways and the dilution process in the model, which could affect the prediction of OH radicals and $N_2O_5$ and thereby the production of $HNO_3$. However, sensitive analyses for various parameters show that the current parameterization of the heterogeneous HONO formation and dilution process in the model allows for robust quantitative constraints ... (see Section S5 and Figures S7, S8 for more details)."

[Figure]

Figure R1 Sensitivity of $N_2O_5$ and OH radical concentrations, production rates of $HNO_3$ from different pathways, as well as their contributions to the $HNO_3$ production to the variations in the value of dilution lifetime from 5 hours to 36 hours in the model. The chosen pollution episode was from 26 to 31 December, 2019. In the base case, a typical dilution lifetime of 24 hour was assumed.

Line 338: The sentence of "The nitrate formation mechanism is different during the different time of a day" is a wrong statement, as the chemical mechanism should be basically the same throughout the day while the dominant formation pathway could change. It should be rephrased or deleted since it is closed to following sentence.

Response: Thanks for the reviewer's comment. We have rephrased this sentence as "The dominant nitrate formation pathway is different during the different time of a day."

Line 340~350: There are two major problems on the evaluation of nighttime nitrate formation pathway. First, the concentration of particulate nitrate observed during nighttime is composed of both daytime remainder and nighttime formation, as it is a long lifetime species. Thus the positive correlation of particulate nitrate concentration with $[NO_2]^2 \times O_3$ might fail to represent the contribution from $N_2O_5$ uptake pathway. Second, what is the time resolution of data points showed in Figure 6? If it is one hour, the level of $[NO_2]^2 \times O_3$ at the point just after sunset, when nighttime formation of nitrate starts, should be the highest over the night under a stable condition without transports. The positive correlation tends to unreasonable accordingly. Suggest replacing the point-to-point correlation with nighttime averages correlation. Similar problems also apply to the daytime cases.

Response: We appreciate the reviewer's suggestion. The time resolution of data points shown in Figures 6 and 7 was one hour. We have replaced the point-to-point correlation with nighttime or daytime average correlation in these two figures (see Figures R2 and R3 below) in the revised manuscript. In addition, to reduce the influences of daytime or nighttime remainder on the analysis of nighttime or daytime nitrate formation, only the data with an obvious peak or increasing trend during the nighttime or daytime were included in the plots.

[Figure]

Figure R2 Nighttime average particulate nitrate concentration (empty circles) as a function of $[NO_2]^2\times[O_3]$ at (a, c) Qingpu and (b, d) Pudong sites in 2018 and 2019. The circles are colored according to aerosol pH and their size is linearly scaled with square root of ALWC. The blue filled circles represent the average of nitrate concentration within a certain $[NO_2]^2\times[O_3]$ interval.

[Figure]

Figure R3 Daytime average particulate nitrate concentration as a function of $O_x$ at (a, c) Qingpu and (b, d) Pudong sites in 2018 and 2019. The circles are colored according to aerosol pH and their size is linearly scaled with square root of ALWC. The blue filled circles represent the average of nitrate concentration within a certain $O_x$ interval. The data points inside the black circle in (c) correspond to low $O_x$ levels but high ALWC and nitrate concentrations.

Line 473~474: References as to the statement that reduction of $NO_2$ could result in the increase of $O_3$ and OH radical are suggested to be provided here.

Response: We have added the relevant references to this statement in the revised manuscript.

Line 522~523: Please explain why regional transport with more aged air plume leads to higher NOR and SOR values before the epidemic periods than that during the epidemic periods? It seems confusing to readers.

Response: The Qingpu site was more easily influenced by the transport of air pollutants from Jiangsu, which is usually more polluted than Shanghai. Before the epidemic, the transport of aged air plume with relatively high nitrate and sulfate concentration from Jiangsu would result in a relatively high NOR and SOR values at the Qingpu site. However, during the epidemic, the emission reduction not only happened in Shanghai, but also in the surrounding areas. As a result, the nitrate and sulfate concentration in the aged air plume from Jiangsu would decrease significantly, leading to a lower NOR and SOR during the epidemic at the Qingpu site.

We have revised manuscript to explain the reason more explicitly.

"In addition, before the epidemic, the transport of aged air plume with relatively high nitrate and sulfate concentrations from upwind regions resulted in relatively high NOR and SOR values at the Qingpu site. However, during the epidemic, the significant decrease in nitrate and sulfate concentrations in the aged air plume due to regional emission reductions leaded to lower NOR and SOR at this site."

References:

Liu, Y. C., Zhang, Y. S., Lian, C. F., Yan, C., Feng, Z. M., Zheng, F. X., Fan, X. L., Chen, Y., Wang, W. G., Chu, B. W., Wang, Y. H., Cai, J., Du, W., Daellenbach, K. R., Kangasluoma, J., Bianchi, F., Kujansuu, J., Petaja, T., Wang, X. F., Hu, B., Wang, Y. S., Ge, M. F., He, H., and Kulmala, M.: The promotion effect of nitrous acid on aerosol formation in wintertime in Beijing: the possible contribution of traffic-related emissions, Atmos. Chem. Phys., 20, 13023-13040, doi: 10.5194/acp-20-13023-2020, 2020.

McDuffie, E. E., Womack, C. C., Fibiger, D. L., Dube, W. P., Franchin, A., Middlebrook, A. M., Goldberger, L., Lee, B., Thornton, J. A., Moravek, A., Murphy, J. G., Baasandorj, M., and Brown, S. S.: On the contribution of nocturnal heterogeneous reactive nitrogen chemistry to particulate matter formation during wintertime pollution events in Northern Utah, Atmos. Chem. Phys., 19, 9287-9308, doi: 10.5194/acp-19-9287-2019, 2019.

Romer, P. S., Wooldridge, P. J., Crounse, J. D., Kim, M. J., Wennberg, P. O., Dibb, J. E., Scheuer, E., Blake, D. R., Meinardi, S., Brosius, A. L., Thames, A. B., Miller, D. O., Brune, W. H., Hall, S. R., Ryerson, T. B., and Cohen, R. C.: Constraints on Aerosol Nitrate Photolysis as a Potential Source of HONO and NOx, Environ. Sci. Technol., 52, 13738-13746, doi: 10.1021/acs.est.8b03861, 2018.

We are grateful to the reviewer for the thoughtful comments on the manuscript. Our point-to-point responses to each comment are as follows (reviewer's comments are in black font and our responses are in blue font).

Comments:

Zang et al., present a comprehensive study to identify the major nitrate formation pathways and their key controlling factors during the winter haze pollution period in the eastern YRD, China using two-year (2018-2019) field observations and detailed observation-constrained model simulations. They find that high atmospheric oxidation capacity is the reason for the winter nitrate aerosol pollution in YRD region in China. And $N_2O_5$ uptake contributes 60-70% in urban and suburban sites in polluted days. The analysis of the observation data is sound, I only have some comments to the model simulations.

Major Issues:

Line 24-27, The quantification of nitrate formation importance is derived from pollution episodes only. The campaign average result should be much more different. Please clarify it.

Response: Thanks for the reviewer's comment. In this study, we focused on the contribution of different processes to nitrate formation during the haze pollution episodes. To be more precise, we have revised this description as "We find that…, with contribution percentages of 69% and 29% in urban areas and 63% and 35% in suburban areas during the haze pollution episodes, respectively." (changes underlined).

The model includes the dry deposition of $HNO_3$, it seems that the authors want to simulate the variation the particle nitrate. I am very interesting whether the modelled nitrate comparable with the observation. Is it possible to provide more details about the intercomparison? In addition, when calculating the contribution of nitrate formation, are you just accumulate the nitrate production rate during a certain period from different channel? Are the only represent the formation potential without considering the dry deposition, what is the role of the dry deposition in the model simulation since it cannot influence any result in the paper?

Response: In this study, we simulated the formation rate (i.e., formation potential) of $HNO_3$ from different pathways but not the concentration of particulate nitrate. Accordingly, the contribution of nitrate formation was the accumulation of the $HNO_3$ production rate from different channel over a certain period (e.g., daytime or nighttime). In the manuscript, we have compared the increasing rates of particulate nitrate with the formation rates of $HNO_3$ for several typical episodes and found that the two rates were comparable. The dry deposition did not influence the formation potential of $HNO_3$, so we have removed its calculation in Section 2.3 of the main text.

The heterogeneous chemistry is well considered in the model simulation, such as the $N_2O_5$ and $NO_2$ uptake mechanism, but limited by the observation, the importance of these reactions cannot be confirmed, If the field measurement of $N_2O_5$ or $ClNO_2$ are available, the result would be more insightful with smaller uncertainties. Here, I suggest the author provide more information about the parameterized $N_2O_5$ uptake and $ClNO_2$ yield in the main text or SI, which could help people to connect the further observation studies that quantifying $N_2O_5$ uptake coefficient and/or $ClNO_2$ yield.

Response: We certainly agree that simultaneous measurements of $N_2O_5$ and $ClNO_2$ would provide strong constraints on the nitrate formation chemistry, but unfortunately such measurements are not available in this study. Instead, we carefully parameterized the heterogeneous nitrate formation pathways based on recent advances on the reaction kinetics and well-measured aerosol data. As mentioned in our replies to the previous comment, the modelled $HNO_3$ production rates were comparable to the measured increasing rates of particulate nitrate during several pollution episodes, indicating our model results are reliable.

According to the reviewer' suggestion, we have added more information about the parameterized $N_2O_5$ uptake and $ClNO_2$ yield in Section 2.3 of the revised manuscript (changes underlined).

"For the heterogeneous hydrolysis of $N_2O_5$, the $N_2O_5$ molecules accommodated on aqueous aerosols can undergo reversible hydrolysis to form $NO_3^-$ and $H_2ONO_2^+$ (R1), followed by the reaction of $H_2ONO_2^+$ with $H_2O$ or $Cl^-$ to form $HNO_3$ and $ClNO_2$ (R2 and R3) (Finlayson-Pitts et al., 1989; Schweitzer et al., 1998; Thornton and Abbatt, 2005):

$$N_2O_5\,(aq) + H_2O\,(l) \underset{k_{1b}}{\overset{k_{1f}}{\rightleftharpoons}} H_2ONO_2^+\,(aq) + NO_3^-\,(aq) \quad (R1)$$

$$H_2ONO_2^+\,(aq) + H_2O\,(l) \xrightarrow{k_2} H_3O^+\,(aq) + HNO_3\,(aq) \quad\quad (R2)$$

$$H_2ONO_2^+\,(aq) + Cl^-\,(aq) \xrightarrow{k_3} ClNO_2\,(g) + H_2O\,(l) \quad\quad (R3)$$

…

$$\gamma N_2O_5 = \frac{4}{c}\,\frac{V_a}{S_a}\,K_H \times k_{1f}\,\times (1 - \frac{1}{\left(\frac{k_2}{k_{1b}} \times \frac{[H_2O]}{[NO_3^-]}\right) + 1 + \left(\frac{k_3}{k_{1b}} \times \frac{[Cl^-]}{[NO_3^-]}\right)}) \quad (3)$$

where $V_a$ is the measured aerosol volume concentration; $K_H$ is the Henry's law coefficient of $N_2O_5$, with a value of 51 M atm$^{-1}$ (Bertram and Thornton, 2009); $k_{1f}$ is the second-order reaction rate constant of $N_2O_5$ with water, which was calculated using a linear function with [H$_2$O], as $3.0 \times 10^4 \times$ [H$_2$O] (Yu et al., 2020a); $\frac{k_2}{k_{1b}}$ and $\frac{k_3}{k_{1b}}$ are the relative rates of reactions of $H_2ONO_2^+$ (aq) with H$_2$O or Cl$^-$ (R2 and R3) versus that with NO$_3^-$ (the reverse reaction of R1), with values determined to be 0.033 and 3.4, respectively (Yu et al., 2020a); and [H$_2$O], [NO$_3^-$], and [Cl$^-$] are the molarity of water, nitrate, and chloride in aerosol, respectively.

…

$$\Phi_{HNO3} = 1 - 1/(1 + \frac{[H_2O]}{\frac{k_3}{k_2} \times [Cl^-]}) \quad\quad\quad (4)$$

where $\frac{k_3}{k_2}$ is the ratio of reaction rates of R3 versus R2, which has been determined to be 105 (Bertram and Thornton, 2009; Yu et al., 2020)."

Monoterpene is very reactive to NO$_3$ radical, and we notice that monoterpene was not included in the model simulation, although the monoterpenes concentration may be low during the winter due to low temperature, but it maybe still have large contribution to the NO$_3$ loss and affect the budget, I encourage the authors do some sensitivity tests to assess the impacts to N$_2$O$_5$ uptake and following nitrate formation.

Response: We appreciate the reviewer's point. We have conducted a sensitivity test for monoterpenes to evaluate their influence on the HNO$_3$ formation. It should be noted that we only have the observation data of monoterpenes obtained using a proton transfer reaction time-of-flight mass spectrometry (PTR-ToF-MS, Vocus, Tofwerk) at an urban site in Shanghai in early November, 2019. We selected the data on 9 November as the ambient temperature (average: 13.3 °C) that strongly affects monoterpene emissions (Guenther et al., 2012), was relatively low on this day, close to the temperature in the winter. The wind speed (average: 0.76 m s$^{-1}$) was also low on this day, which limits the transport and dilution of monoterpene emissions. The monoterpene concentration on this day ranges from 0.009 ppb to 0.070 ppb, with an average of 0.038 ppb. The sensitivity analysis shows that when the monoterpene chemistry was considered, the N$_2$O$_5$ concentration and HNO$_3$ production rate from N$_2$O$_5$ hydrolysis (pHNO$_{3(N2O5)}$) both had a decrease, especially during the nighttime with high N$_2$O$_5$ concentration (Figure S9a, b). However, such decrease was relatively small; the average N$_2$O$_5$ concentration and pHNO$_{3(N2O5)}$ decreased by 23% and 12% during the nighttime, respectively. In addition, the contribution of heterogeneous N$_2$O$_5$ hydrolysis to HNO$_3$ formation only decreased by 2.7% (Figure S9c). Notably, the average temperature in the selected winter haze episode was 8.1 °C, which was significantly lower than the temperature on 9 November, so the concentration of monoterpenes should be smaller, as is their impact on the HNO$_3$ formation. To sum up, the low monoterpene emissions had no significant impact on the budget of NO$_3$ radicals and N$_2$O$_5$ as well as the formation of HNO$_3$ during the winter haze pollution episodes in eastern YRD.

In the revised manuscript, we have added the above sensitivity analyses and Figure R1 to the supplement as a new section (Section S6. Potential influences of monoterpenes on HNO$_3$ production). In addition, we have added the following discussion to Section 3.4 of the main text.

"In addition, monoterpenes that are very reactive to NO$_3$ radicals (Atkinson and Arey, 2003) were not included in the model, because their measurements are not available in this study. However, a case study considering the monoterpene chemistry in the model shows that the low monoterpene emissions during the winter did not significantly affect the budget of $NO_3$ radical and $N_2O_5$ and thereby the nighttime $HNO_3$ production (see Section S6 and Figure S9 for more details)."

[Figure]

Figure R1 Sensitivity of $N_2O_5$ concentration, production rates of $HNO_3$ from $N_2O_5$ hydrolysis ($pHNO_{3(N2O5)}$), as well as its contribution to the $HNO_3$ formation ($pHNO_{3(N2O5)}$/ $pHNO_{3(total)}$), to the inclusion of monoterpenes in the model simulation. The chosen episode was from 26 to 31 December, 2019. The base case did not consider the effect of monoterpenes.

Line 249-250, why only constrain the sum NO and $NO_2$, if the NO and $NO_2$ not constrained separately but only the sum, I guess the modeled nocturnal NO always be zero when $O_3$ over ppb. While in fact NO spikes by local emission always observed in urban regions during the nighttime, which would lead to a bias of nitrate formation from $N_2O_5$ uptake (possibly an overestimation).

Response: Thanks for the reviewer's comment. We have tried to constrain NO and $NO_2$ separately in the model, but when we did this, the simulated nighttime concentrations of $NO_3$ radical and $N_2O_5$ were extraordinarily low during the whole observation period, owing to the titration of $NO_3$ by NO. In addition, high $N_2O_5$ peaks were simulated during the daytime likely due to the high $O_3$ concentrations in the model, which is unreasonable.

Therefore, we constrained the sum NO and $NO_2$ but let their specific ratios be simulated by the model. As shown in Figure 8 in the main text and Figure S5 in the supplement, the simulated $NO_2$ concentration was generally in good agreement with the observation, which would also be the case for NO given that the sum of NO and $NO_2$ was constrained by observation. The NO spikes did exist during the nighttime in some episodes, which could lead to an overestimation of $NO_2$. However, as discussed in the manuscript, as the $O_3$ concentration in the model was constrained by the observation, which was very low (below 5 ppb) during the NO spikes periods, the overestimation of $NO_2$ did not significantly affect the prediction of $N_2O_5$.

Figure 8 case 1, the observed $NO_2$ during daytime and nighttime had a lower and higher biases, are they mean the modelled nitrate during the daytime is lower and nighttime is higher. This phenomenon also happened in case 3.

Response: The overestimation of $NO_2$ during the nighttime was due to the NO spikes. As we explained in the previous comment, the $O_3$ concentration in the model was constrained by the measured value, which was very low, the overestimation of $NO_2$ did not significantly affect the modelled $N_2O_5$ and its contribution to $HNO_3$ formation. The bias of modelled $NO_2$ during the daytime is quite small compared to that during the nighttime, it therefore might also have no significant impact on the model results.

Line 81 or change to "and"

Response: We have revised this.

Line 361 weaker change to "weak".

Response: We have revised this.

Line 346 the value 15000 misses the unit, may be $pppbv^3$.

Response: We have added the unit $ppb^3$ for the value.

[revised manuscript text omitted]

Considering the uncertainty in the dilution rate constant ($k_{dil}$), we also performed a sensitivity analysis for $k_{dil}$ by varying its value from 0.028 h$^{-1}$ to 0.2 h$^{-1}$ (corresponding to a dilution lifetime of 5 hours to 36 hours) to evaluate its influence on HNO$_3$ production in a typical pollution episode at the Pudong site (see Figure S8). As the dilution lifetime varied from 5 hours to 36 hours, the average concentrations of N$_2$O$_5$ and OH radicals changed within -23%/+0.8% and -21.6%/+10.8%, respectively (Figure S8a, d), compared to the base case (dilution lifetime: 24 hours) during the episode. Accordingly, the HNO$_3$ production rates from the heterogeneous hydrolysis of N$_2$O$_5$ and gas-phase OH + NO$_2$ reactions changed within -17%/+1.2% and -33%/+12% (Figure S8b, e) and the relative contributions of the two pathways changed within -2.5%/+5.5% and -5%/+2.3% (Figure S8c, f), respectively. The relatively small changes in the rates and relative contributions of the two HNO$_3$ production pathways upon variations in $k_{dil}$ from 0.028 h$^{-1}$ to 0.2 h$^{-1}$ suggest that the simplified parameterization of the dilution process using a constant $k_{dil}$ would not result in significant uncertainty in the model results.

**S6. Influence of monoterpenes on HNO$_3$ production**

The consumption of $NO_3$ radicals by monoterpenes during nighttime can influence the budget of $NO_3$ radicals and $N_2O_5$ and thereby the formation of $HNO_3$. We have conducted a sensitivity test for monoterpenes to evaluate their influence on the $HNO_3$ formation. It should be noted that we only have the observation data of monoterpenes obtained using a proton transfer reaction time-of-flight mass spectrometry (PTR-ToF-MS, Vocus, Tofwerk) at an urban site in Shanghai in early November, 2019. We selected the data on 9 November as the ambient temperature (which strongly affects monoterpene emissions) on this day was relatively low (average: 13.3 °C), close to the temperature in winter. The wind speed was also low (average: 0.76 m s$^{-1}$) on this day, which limits the transport and dilution of monoterpene emissions. The monoterpene concentration on this day ranges from 0.009 ppb to 0.070 ppb, with an average of 0.038 ppb. The sensitivity analysis shows that when the monoterpene chemistry was considered, the $N_2O_5$ concentration and $HNO_3$ production rate from $N_2O_5$ hydrolysis (pHNO$_{3(N2O5)}$) both had a decrease, especially during the nighttime with high $N_2O_5$ concentration (Figure S9a, b). However, such decrease was relatively small; the average $N_2O_5$ concentration and pHNO$_{3(N2O5)}$ decreased by 23% and 12% during the nighttime, respectively. In addition, the contribution of heterogeneous $N_2O_5$ hydrolysis to $HNO_3$ formation only decreased by 2.7% (Figure S9c). Notably, the average temperature in the selected winter haze episode was 8.1 °C, which was significantly lower than the temperature on 9 November, so the concentration of monoterpenes should be smaller, as is their impact on the $HNO_3$ formation.

The above analyses suggest that the low monoterpene emissions had no significant impact on the budget of $NO_3$ radicals and $N_2O_5$ as well as the formation of $HNO_3$ during the winter haze pollution episodes in eastern YRD.

[Figure]

Figure S1 (a) surface area and (b) volume concentrations of dry PM$_{2.5}$ as a function of PM$_{2.5}$ mass concentration at the Qingpu site in 2019.

[Figure]

Figure S2 Time series of temperature, relative humidity (RH), aerosol liquid water content (ALWC), NO$_x$, O$_3$, O$_x$, nitrogen oxidation ratio (NOR), as well as PM$_{2.5}$ and major particulate compositions at the Qingpu site in winter 2019.

[Figure]

Figure S3 Correlation between the concentrations of PM$_{2.5}$ and nitrate, sulfate and ammonium.

[Figure]

Figure S4 Frequency distribution of εHNO$_3$ under different PM$_{2.5}$ pollution conditions at (a-c) Qingpu and (d-f) Pudong sites during winter 2019.

[Figure]

Figure S5 Time series of particulate nitrate, $NO_2$, $O_x$, ALWC, OH, $N_2O_5$, as well as the formation rates of $HNO_3$ from different processes during the two selected pollution episodes at the Qingpu site in 2019. The simulated data with RH > 95% were not included in the figure.

[Figure]

Figure S6 Average diurnal profile of $HNO_3$ production rates from the heterogeneous and gas-phase processes during all the six pollution episodes at (a) Qingpu and (b) Pudong sites.

[Figure]

Figure S7 Sensitivity of (a, d) HONO concentration and production rates of (b, e) HONO and (c, f) HNO₃ to the variations in the values of key parameters of the heterogeneous HONO formation pathways in the model. Episode 1 (a-c) was from 26 to 31 December, 2019. Episode 2 (d-f) was from 11 to 15 January, 2020. The base case was simulated using the best guess of the parameters as listed in Table 1 in the main text.

[Figure]

Figure S8 Sensitivity of N₂O₅ and OH radical concentrations, production rates of HNO₃ from different pathways, as well as their contributions to the HNO₃ production to the variations in the value of dilution lifetime from 5 hours to 36 hours in the model. The chosen pollution episode was from 26 to 31 December, 2019. In the base case, a typical dilution lifetime of 24 hours was assumed.

[Figure]

Figure S9 Sensitivity of $N_2O_5$ concentration, production rates of $HNO_3$ from $N_2O_5$ hydrolysis ($pHNO_{3(N2O5)}$), as well as its contribution to the $HNO_3$ formation ($pHNO_{3(N2O5)}$/ $pHNO_{3(total)}$), to the inclusion of monoterpenes in the model simulation. The chosen episode was from 26 to 31 December, 2019. The base case did not consider the effect of monoterpenes.

[revised manuscript text omitted]

---

## Author Response (AR2)

Comments to the author:

The revised manuscript has been reviewed again by Referee #2 who acknowledges the revisions to be adequate. Following the recommendations of both referees, I am pleased to accept the paper for publication in ACP subject to technical corrections. See one technical comment by Referee #2. In addition, please specify units for $k_{dep}$ and PBL in Table 1.

Response: Thanks to the editor for handling our manuscript. We have added the units for $k_{dep}$ and PBL in Table 1 of the revised manuscript.

Response to the reviewer

Line 648, JA and CH provided the $NO_x$ emission inventory, while it seems that $NO_x$ emission inventory not used in the studies.

Response: Thanks for the reviewer's comment. We used the $NO_x$ emission inventory and an empirical emission ratio of HONO to $NO_x$ to get the vehicular emissions of HONO in the model simulation. We included a statement in the supplement of the previous version of manuscript. To make it clearer, we have moved this statement to the main text of the revised manuscript (see below).

"We also considered the direct emissions of HONO from vehicles based on a 4 km × 4 km emission inventory of $NO_x$ and an empirical emission ratio (0.8%) of HONO to $NO_x$ (Kurtenbach et al., 2001; An et al., 2021)."

The authors provide a detailed response and careful revision of the manuscript. I would like to recommend it. In addition, I want to highlight that the level of $N_2O_5$ is very sensitive to NO, so the constrain of sum $NO_x$ rather than NO and $NO_2$ maybe under estimate nighttime NO, which lead to the modelled $N_2O_5$ concentration and its contribution to nitrate represent an upper limit to some extent. This problem cannot be resolved in the work but should take into consideration in future simulations.

Response: We agree that there might be an underestimation of NO during the nighttime when the NO concentration was high. However, we can see from the case study that the $O_3$ concentration was very low during the NO spikes periods, thus the overestimation of $N_2O_5$ concentration and its contribution to nitrate might not be significant. We will try to find a better solution to constrain the $N_2O_5$ simulation in our future studies.

[revised manuscript text omitted]